

# Observational Constraints Suggest a Smaller Effective Radiative Forcing from Aerosol-Cloud Interactions

Chanyoung Park[1]*, Brian J. Soden[1], Ryan J. Kramer[2], Tristan S. L'Ecuyer[3], Haozhe He[4]

1. Rosenstiel School of Marine, Atmospheric, and Earth Science, University of Miami, Miami, FL, USA,

2. NOAA/Geophysical Fluid Dynamics Laboratory, Princeton, NJ, USA,

3. University of Wisconsin–Madison, Madison, WI, USA,

4. High Meadows Environmental Institute, Princeton University, Princeton, NJ, USA.

*Corresponding author: Chanyoung Park (chanyoung.park@miami.edu)

## Abstract

The effective radiative forcing due to aerosol-cloud interactions (ERFaci) is difficult to quantify, leading to large uncertainties in model projections of historical forcing and climate sensitivity. In this study, satellite observations and reanalysis data are used to examine the low-level cloud radiative responses to aerosols. While some studies it is assumed that the activation rate of cloud droplet number concentration ($N_d$) in response to variations in sulfate aerosols ($SO_4$) or the aerosol index (AI) has a one-to-one relationship in the estimation of ERFaci, we find this assumption to be incorrect, and demonstrate that explicitly accounting for the activation rate is crucial for accurate ERFaci estimation. This is corroborated through a "perfect-model" cross validation using state-of-the-art climate models, which compares our estimates with the "true" ERFaci. Our results suggest a smaller and less uncertain value of the global ERFaci than previous studies (-0.39 ± 0.29 W m$^{-2}$ for $SO_4$ and -0.24 ± 0.18 W m$^{-2}$ for AI, 90% confidence), indicating that ERFaci may be less impactful than previously thought. Our results are also consistent with observationally constrained estimates of total cloud feedback and "top-down" estimates that models with weaker ERFaci better match the observed hemispheric warming asymmetry over the historical period.

## 1. Introduction

Anthropogenic aerosols impact the Earth's radiation balance at the top of the atmosphere and alter cloud properties over the industrial era (Boucher et al., 2013; Raghuraman et al., 2021; Kramer et al., 2021). They directly alter the radiation budget by scattering and absorbing solar radiation and indirectly influence it by serving as cloud condensation nuclei (CCN), which



modifies cloud properties and can extend their duration. This increase in aerosol concentration
leads to smaller cloud droplets and higher cloud albedos, known as the "Twomey effect" (e.g.,
Twomey, 1977), enhancing the radiative forcing due to aerosol-cloud interactions (RFaci).
Additionally, aerosols affect cloud microphysical properties (e.g., Albrecht, 1989; Pincus and
Baker, 1994), such as reducing precipitation, which increases cloud liquid water path (LWP),
lifetime, and fraction, a process termed cloud adjustment (CA). Thus, together, RFaci and CA
are intrinsically interconnected through the cloud droplets (Mülmenstädt and Feingold, 2018),
and constitute the ERFaci, which is highly uncertain and often larger than the direct radiative
impact of aerosols (Forster et al., 2007; Zelinka et al., 2014; Smith et al., 2020a).

Estimating the ERFaci, especially in low-level clouds which are the dominant contributor of
aerosol-cloud interactions to ERFaci (Christensen et al., 2016; Bellouin et al., 2020; Forster et
al., 2021), is critical for accurately identifying cloud feedback mechanisms and determining
climate sensitivity (Rosenfeld, 2006; Boucher et al., 2013; Sherwood et al., 2020). Our study
provides quantitative insights into the ERFaci using both satellite observations and reanalysis
data. A key component of our analysis is the activation rate, which serves as a metric for
assessing the actual impact of aerosols on cloud droplet number concentrations. The
conventional assumption is that the activation rate has a one-to-one relationship when aerosols
convert into cloud droplets and is typically not explicitly incorporated into the estimation
process of ERFaci. Our results suggest the importance of considering the activation rate when
evaluating the interactions between aerosols and clouds. To evaluate the robustness of our
results, we conduct a "perfect-model" cross validation using Coupled Model Intercomparison
Project Phase 6 (CMIP6) simulations. This form of cross-validation is widely used in statistics
and machine learning to assess the generalizability of predictive models and prevent overfitting
(Wenzel et al., 2016; Knutti et al., 2017; Brunner et al., 2020). Through this approach we
demonstrate that explicitly including the activation rate is essential to improving the accuracy
of ERFaci estimates. Although open questions remain, the cross-validation clearly
demonstrates the improved predictive skill of our model and thus increases the confidence of
our estimates of ERFaci.

In the main text, our analysis primarily focuses on $SO_4$ as an aerosol proxy, recognized as a
major contributor among other aerosol types such as black carbon, organic carbon, sea salt, and
dust (Charlson et al., 1992; McCoy et al., 2018). However, results derived from the Aerosol





Index (AI), a more generalized aerosol metric (e.g. Douglas and L'Ecuyer 2019, 2020), also
show a high degree of consistency.

## 2. Results

### 2.1 Activation Rate

Some approaches to estimate the ERFaci with aerosol concentrations have operated under a
key assumption: the natural logarithm of aerosol concentration correlates proportionally with
the natural logarithm of cloud droplet number concentration (Boucher and Lohmann, 1995;
Wall et al., 2022, 2023). This ratio, commonly referred to as the activation rate, quantifies the
efficiency with which aerosol particles convert into cloud droplets. The hypothesized cause-
effect relationship between aerosols and clouds is important to understand and to be dealt in
the process of aerosol-cloud interactions, as it involves an increase in CCN leading to an
increase in $N_d$, which subsequently influences cloud properties. To verify the key assumption,
we performed a linear regression. As illustrated in Fig. 1, the regression coefficients between
$\ln(N_d)$ and $\ln(SO_4)$ were calculated. Our results show that, in most regions, these coefficients
are positive but less than 1. This indicates that while there is a proportional relationship, it is
not a one-to-one increase; rather, the activation rate varies across different geographic locations.
Regions with shallow cumulus clouds, such as the central Pacific, show a notably weaker
$\partial\ln(N_d)/\partial\ln(SO_4)$ coefficient, while areas with stratocumulus clouds, like those off the coasts
of continents, display a relatively stronger positive regression with significant correlation
coefficient (Fig. 1). Repeating our analysis using $\partial\ln(N_d)/\partial\ln(AI)$ also yields results consistent
with those for $\ln(SO_4)$, emphasizing the necessity of addressing this assumption within the
ERFaci estimation process (Fig. A1). The relatively low correlation coefficients observed for
$\partial\ln(N_d)/\partial\ln(AI)$ may be attributed to the use of column-integrated quantities, AI from MODIS,
which do not account for the vertical structure of aerosols. Consequently, they may not
accurately represent aerosol concentrations at cloud base height. In contrast, the use of $SO_4$
concentration at 925 hPa in the analysis provides a more precise representation of CCN
concentrations near the cloud base (Painemal et al., 2017). This leads to a higher linearity
between $SO_4$ and $N_d$, establishing $SO_4$ a more relevant indicator for evaluating the interactions
between aerosols and low-level cloud formation (Fig. 1 vs Fig. A1).





**2.2 Observationally Constrained ERFaci**

To isolate the contributions of different environmental factors to the low cloud radiative effect, we first have employed a cloud controlling factor (CCF) analysis (Scott et al., 2020; Wall et al., 2022) with a particular focus on elucidating the relationship between aerosol concentrations and the low cloud radiative effect. This relationship is known as a susceptibility and constitutes one of the key components in the estimation of ERFaci. Our implementation of the CCF analysis basically follows the method described by Wall et al. (2022) (See more details in Appendix A).

We now proceed to estimate the observationally constrained ERFaci (ERFaci_obs), considering two scenarios: one with and the other without the inclusion of the activation rate. The basic form of ERFaci_obs following Wall et al. (2022), where the activation rate is not explicitly included, can be expressed as follows:

$$\text{ERFaci\_obs} \approx \sum_{k=1}^{10} \left( \frac{\partial \text{CRE\_lcld}}{\partial \ln(Y)} \right)_k W_k \times \Delta \ln(Y), \qquad (1)$$

where CRE_lcld represents the cloud radiative effect from low-level clouds, Y represents either $SO_4$ or AI, and $W_k$ represents the fraction of LWP in state k ($W_k = \frac{\text{number in LWP state k}}{\text{total number}}$). The right-hand-side of the equation consists of two main parts: one is the susceptibility of the low-cloud radiative effect to variations in aerosol concentrations, which can be derived from CCF analysis using observations and the other one is the changes in aerosol concentrations from pre-industrial (PI) to present-day (PD). Due to the lack of observational data on PI aerosol concentrations, we employ the outputs of CMIP6 historical experiments. As expected, changes in $SO_4$ concentrations exhibit distinctive spatial patterns characterized by interhemispheric asymmetry, with particularly large values in proximity to major industrial regions on the Eurasian and North American continents (Fig. 2a).

In light of Fig. 1, the basic form of ERFaci_obs in equation (1) can be expanded to incorporate the influence of the activation rate by accounting for the interactions between aerosols and cloud droplet formation. This modified equation can be expressed as follows:



$$\text{ERFaci\_obs} \approx \sum_{k=1}^{10} \left( \frac{\partial \text{CRE\_lcld}}{\partial \ln(N_d)} \times \frac{\partial \ln(N_d)}{\partial \ln(Y)} \right)_k W_k \times \Delta \ln(Y), \qquad (2)$$


where the low cloud susceptibility is now the product of two terms: The susceptibility of low
cloud CRE to $N_d$ and the activation rate of Y to $N_d$.

Our analysis reveals pronounced differences in susceptibility in how low cloud radiative effects
respond to variations in aerosol concentrations across the globe depending on whether
activation rate is considered or not. The inclusion of the activation rate in our analysis
significantly diminishes the sensitivity of clouds to aerosols (Fig. 2b vs Fig. 2c). Noticeable
decreases in susceptibility are captured in mid-latitudes and in subtropical regions where low
clouds are dominant. This also indicates that the $\partial \ln(\text{CRE\_lcld})/\partial \ln(SO_4)$ correlation without
activation rate is partially attributable to factors other than the $N_d$-mediated mechanism (Wood
et al., 2012; Gryspeerdt et al., 2016; Gryspeerdt et al., 2019).

Both methods of estimating ERFaci_obs show that an increase in aerosol concentration
correlates with a negative cloud radiative adjustment that is especially prevalent in areas
dominated by low clouds (Fig. 2d,e). However, due to the reduced susceptibility, the estimated
ERFaci_obs is significantly smaller when activation is explicitly accounted for (Fig. 2e) than
when it is not (Fig. 2d). The global ERFaci_obs is ~50% smaller with activation (-0.39 W m$^{-2}$)
than without (-0.79 W m$^{-2}$). Similar results are obtained if one uses AI instead of $SO_4$ as the
measure of aerosol concentration (Fig. A2d,e). These results highlight the sensitivity of this
approach to explicit consideration of the activation rate.

**2.3 Perfect-Model Cross Validation**
In this section, we perform a "perfect-model" cross validation exclusively using CMIP6
simulations to assess which of the two approaches—considering activation rate or not—is more
accurate. Specifically, each model from single-forcing (aerosol-only) experiments is
sequentially treated as the "truth" with its ERFaci considered the "true" value. Meanwhile, the
same model from historical simulations, assumed to be a pseudo-observation, estimates ERFaci
for comparison with the "true" ERFaci. The resulting root mean-square error (RMSE) provides
a quantitative measure of the accuracy of the ERFaci estimates.




As an initial step in the "perfect-model" test, single-forcing (aerosol-only) CMIP6 simulations
are used to establish the true ERFaci for each model, referred to as ERFaci_true, which
provides a benchmark for assessing the accuracy of the ERFaci estimated from the monthly
outputs of CMIP6 historical experiments using equations (1) and (2), where the model is treated
as a pseudo-observation and the estimate is referred to as ERFaci_est. Because the number of
CMIP6 models that provide single-forcing (aerosol-only) simulations for ERFaci_true is
limited, we also explore another technique for estimating ERFaci introduced by Soden and
Chung (2017; referred to as ERFaci_SC17) that has been previously shown to agree well with
ERFaci_true (Chung and Soden, 2017). For more details on the estimation of these three
different ERFaci using CMIP6 model outputs, please refer to Appendix A. A comparison, for
the "perfect-model" test, of ERFaci_est with both ERFaci_true and ERFaci_SC17 is provided
below.

Fig. 3 illustrates the correlation between ERFaci_true and two alternative approaches derived
from CMIP6 model output. The estimates of ERFaci_est that omit the activation rate fail to
replicate the "true" ERFaci values accurately, with RMSE of 0.68 W m$^{-2}$ and bias of 0.56 W m$^{-2}$.
Conversely, incorporating an explicit activation rate into the ERFaci estimates provides
significantly better agreement with ERFaci_true, reducing both the RMSE and bias by around
40% (Fig. 3a).

ERFaci_SC17 exhibits the best agreement with ERFaci_true, with significantly smaller RMSE
(0.14 W m$^{-2}$) and bias (0.1 W m$^{-2}$) (Fig. 3b). This consistency allows us to expand the sample
size of CMIP6 models, with which we can evaluate ERFaci_est by using ERFaci_SC17 as a
surrogate for ERFaci_true (Fig. 3c). This expanded cross-validation once again highlights the
importance of including the activation rate in ERFaci estimates, as it reduces both the RMSE
and bias in ERFaci_est by around 45%. Substituting AI for $SO_4$ in the calculation of ERFaci_est
yields similar results, which reduces RMSE more than 40%, emphasizing the importance of
explicitly including activation rate (Fig. A3). Our "perfect-model" cross validation analysis
with idealized model experiments from CMIP6 leads us to conclude that the inclusion of the
activation rate is essential for accurate estimates of ERFaci.





### 2.4 Comparison with previous ERFaci estimates

Now, we compare our observationally constrained estimates of ERFaci_obs with those previously estimated. Our global estimates with inclusion of activation rate yield an ERFaci of $-0.39 \pm 0.29$ W m$^{-2}$ for SO$_4$ and $-0.24 \pm 0.18$ W m$^{-2}$ for AI (Fig. 4). These values are at the lower bound when compared with ERFaci values reported in the Sixth Assessment Report of the Intergovernmental Panel on Climate Change (IPCC; Forster et al., 2021) as well as the values proposed by the World Climate Research Program (WCRP; Bellouin et al., 2020). However, it is worth noting that, as the ERFaci from WCRP has a highly skewed distribution with its highest probability occurring around $-0.4$ W m$^{-2}$, which is entirely consistent with our observational estimates (Fig. 4). Given the multiple lines of evidence introduced by the WCRP, which employs a process-oriented approach to bound ERFaci, our estimates offer further evidence to support estimates on the lower end of their range. Furthermore, these constrained ERFaci_obs are also consistent with the "top-down" estimates provided by Wang et al. (2021), which demonstrate that models exhibiting weaker ERFaci are more in line with the observed variations in global mean surface temperature as well as hemispheric warming asymmetry during the historical period.

As we emphasized the significant impact of including the activation rate in the ERFaci estimation process, with this inclusion, the ERFaci_obs values are approximately one-half for SO$_4$ and one-fifth for AI of those estimated without considering the activation rate, respectively ($-0.79 \pm 0.28$ W m$^{-2}$ for SO$_4$ and $-1.14 \pm 0.29$ W m$^{-2}$ for AI).

### 2.5 Implications for Cloud Feedback

Our observational estimate of ERFaci is on the lower end compared to previous estimates. This finding also has implications for our understanding of cloud feedback mechanisms. Following Wang et al. (2021), we compare the CMIP6 historical simulations of ERFaci across different climate models with their corresponding values of total cloud feedback, which are derived from the regression slope of total cloud radiative response to global-mean temperature anomalies from the abrupt-4xCO$_2$ experiment (Fig. 5). For this analysis, we use the ERFaci_SC17 since it ensures the widest possible selection of climate models (Table A1). Among the models we assessed, we identified a subset of 15 that we termed 'GOOD HIST' models (Appendix A). These models are characterized by their small discrepancies in simulating global-mean historical surface warming when compared to the GISTEMP observational data, indicating a



higher reliability in their historical climate simulations. Within this subset, a strong negative
correlation (r = -0.85) exists between ERFaci_SC17 and the total cloud feedback, which is
much more pronounced than in the remaining models (r = -0.31). The strong correlation in the
'GOOD HIST' models highlights the compensation that occurs between historical aerosol
forcing and cloud feedback in order for models to reproduce the observed historical global-
mean temperature.

Also shown are the probability density functions for the observation-based estimates of
ERFaci_obs, taking into account the activation rate, and utilizing both $SO_4$ and the AI.
Alongside, we also consider the observationally constrained estimates of total cloud feedback,
which a recent study (Ceppi and Nowack, 2021) has quantified at $0.43 \pm 0.35$ W m$^{-2}$ K$^{-1}$ (90%
confidence). These distributions help illustrate that our constraints on ERFaci fall within the
realistic bounds of total cloud feedback strength. The best estimates, which show the highest
probability (indicated by stars), also align with those from the 'GOOD HIST' models and
support the validity of our constraints.

## 3. Conclusion

Our study offers critical insights into the quantification of ERFaci, a topic that remains a
significant source of uncertainty in understanding climate sensitivity. By integrating both
satellite observations and reanalysis data with a focus on the activation rate of cloud droplet
number concentration in response to aerosol variations, we provide a more sophisticated
understanding of the impact of aerosols on low-level clouds. Our findings, validated through a
"perfect-model" cross validation using CMIP6 model simulations, reveal a lower global
ERFaci estimate, suggesting that the influence of aerosols, particularly with $SO_4$, on climate
forcing may be less substantial than previously assumed.

## Appendix A: Methods

### A1 Observation and Reanalysis Data

### A1.1 CERES

In this study, we analyze observational datasets characterized by their monthly temporal
resolution and their geographical coverage extending from 50°S to 50°N, with a particular
focus on oceanic regions due to unreliable retrieval over land (Jia et al., 2019; Gryspeerdt et



al., 2022; Jia and Quaas, 2023). The dataset spans from January 2003 through December 2019
and all data fields were interpolated onto a 2.5° × 2.5° grid.

Our analysis employs monthly gridded satellite observations from the CERES FluxByCldTyp
Edition 4.1 dataset, focusing on a combined analysis of cloud fraction and top-of-atmosphere
radiative flux, segmented by cloud optical depth and cloud top pressure (CTP). We categorize
clouds into low (CTP > 680 hPa) and non-low clouds (CTP ≤ 680 hPa) based on their CTP
values. Due to the passive retrieval mechanisms of satellite instruments, the detection of low-
level clouds is notably challenged by the obscuration from upper-level clouds. This limitation
highlights the importance of accurately estimating the fraction of non-obscured or non-
overlapped low-level clouds (Scott et al., 2020). To address this, we define the non-obscured
low-cloud fraction as following equation:

$$L_n = \frac{L}{1 - U} \; , \qquad (A1)$$

where L and U represent the low and non-low cloud fraction retrieved by the satellite, and $L_n$
denotes the total low-level cloud fraction relative to the area of each grid box that is not
obscured by upper-level clouds. With this relationship, we can extend its application to the
cloud radiative effect (CRE) attributable to non-obscured low-level clouds (CRE_lcld). Further
details regarding this equation can be found in the work of Scott et al. (2020).

**A1.2 MERRA-2 reanalysis**

We also use monthly meteorological fields for cloud controlling factor analysis and sulfate
aerosol mass concentrations at 925 hPa derived from the Modern-Era Retrospective analysis
for Research and Applications, Version 2 (MERRA-2) reanalysis (Randles et al., 2017; Gelaro
et al., 2017). MERRA-2 integrates observations with global model simulations to provide
estimates of atmospheric conditions. Specifically for sulfate aerosols, it employs bias-corrected
observations of total aerosol optical depth in conjunction with a comprehensive model
addressing the emissions, removal processes, and chemistry of sulfate and its precursor gases.
The assimilation process adjusts for aerosol hydration in humid conditions and excludes cloud-
adjacent pixels to mitigate retrieval bias. A notable constraint of these data is that, while the
total aerosol optical depth is observationally constrained, the distribution and vertical profiles
of aerosol species are model-derived. Nevertheless, the sulfate concentration estimates exhibit
a strong correlation with independent satellite measurements of cloud droplet number



concentration (McCoy et al., 2018).

**A1.3 MODIS**
We employ the aerosol index (AI) as an alternative proxy for aerosol concentration from the
Moderate Resolution Imaging Spectroradiometer (MODIS) on both the Aqua and Terra
satellites (datasets MYD08_M and MOD08_M, respectively). These two are combined to
enhance the robustness of our analysis. The AI is derived from the product of the Angstrom
exponent and the aerosol optical depth (AOD) at 550 nm. The Angstrom exponent itself is
derived from the wavelength dependency of the AOD, measured at 550 nm and 870 nm,
providing insight into the size distribution of aerosols (i.e. smaller Angstrom exponent suggests
larger particles). Notably, AI has demonstrated a more robust correlation with CCN compared
to the use of AOD alone (Stier, 2016; Gryspeerdt et al., 2017; Hasekamp et al., 2019).

To calculate $N_d$ based on the adiabatic approximation, we use daily gridded $N_d$ estimates from
MODIS (Gryspeerdt et al., 2022) and combine the data from the Aqua and Terra satellites. The
retrievals at 3.7 μm, known to yield more accurate cloud droplet effective radius ($r_e$)
measurements under inhomogeneous conditions, are employed (Zhang and Platnick, 2011). $N_d$
measurements may be subject to biases under specific conditions, such as when the cloud
droplet effective radius is significantly small, when the cloud visible optical thickness is low,
or when three-dimensional radiative transfer effects impact the observed radiances. To enhance
the accuracy and reliability of our $N_d$ retrievals, we implement a rigorous sampling strategy
("BR17 sampling method" in Gryspeerdt et al., 2022). This introduced by Bennartz and Rausch
(2017) demonstrates the highest correlation with aircraft data.

For LWP, MODIS MCD06COSP dataset version 6.2.0 (Pincus et al., 2023) is used. This dataset
represents a combined product derived from both the Aqua and Terra satellites. To accurately
estimate the aerosol indirect effect, it is essential to control variations in LWP, in line with the
foundational assumption of the Twomey effect. In our analysis, we achieve this by categorizing
LWP observations into ten equal bins, each covering a range of 30 g cm$^{-2}$, up to a maximum of
300 g cm$^{-2}$. This categorization is based on the finding that over 99% of our observations do
not exceed 300 g cm$^{-2}$, thus allowing us to maintain LWP within a controlled and effectively
constant range across our dataset.



### A1.4 GISTEMP

The global surface temperature observations used in our analysis are sourced from the GISS Surface Temperature Analysis (GISTEMP v4) (Lenssen et al., 2019). We evaluate how well the models simulate the global-mean historical surface warming by the GOOD HIST index: the absolute difference in global-mean historical warming between CMIP6 models and GISTEMP data (Table A1). The historical warming is defined as the averaged surface temperature in 1990–2014 minus that in 1880–1909. So, the models that are good at simulating the historical warming have a small GOOD HIST index.

### A2 CMIP6 Data

Due to the unavailability of direct observational records for pre-industrial aerosol emissions, we rely on the outputs from historical simulations with realistic emissions of greenhouse gases, aerosols, and aerosol precursor gases conducted by CMIP6 models to estimate changes in aerosol concentration ($\Delta \ln(Y)$, where Y represents either $SO_4$ or AI). The pre-industrial (PI) period was defined as the years 1850 to 1899, and the present-day (PD) period was set from 1965 to 2014, each spanning 50 years to remove interannual variability. In the analysis, 13 models are used for $\Delta \ln(SO_4)$ and 9 models for $\Delta \ln(AI)$, all models of which are among the 21 models that provide ERFaci_true. The specific models used in our analysis are listed in Table A1. It is important to note that, for the CMIP6 models, the emission concentrations of sulfur dioxide, a precursor to $SO_4$, are specified from the Community Emission Data Set (CEDS; Hoesly et al., 2018), and thus the projected changes in $\Delta \ln(SO_4)$ are highly consistent across models. The specified decadal trends in regional sulfate in the models are also consistent with surface observations (Aas et al., 2019).

To evaluate our observationally constrained estimate of the ERFaci (ERFaci_obs), we employed 21 distinct models conducting single-forcing (aerosol-only) experiments (ERFaci_true). These models are from the Radiative Forcing Model Intercomparison Project (RFMIP; Pincus et al., 2016), specifically Tier 1 piClim-control and piClim-aer experiments with prescribed sea surface temperatures (SST) and sea ice derived from a climatology of pre-industrial conditions. These simulations are run for 30 years, incorporating realistic aerosol emissions in 1850 and 2014 to represent PI and PD conditions, respectively. This ensures an accurate estimation of the true baseline of ERFaci resulting solely from aerosol-cloud interactions. We use 30-year time periods for the PI and the PD scenario to evaluate ERFaci.



Consequently, the ERFaci derived from these experiments is referred to as ERFaci_true.

**A3 Cloud Controlling Factor Analysis**
To improve our understanding of the low cloud radiative effect, we have employed a cloud
controlling factor (CCF) analysis (Scott et al., 2020; Wall et al., 2022). This approach allows
us to constrain the physical factors influencing low cloud properties and their subsequent
radiative impacts. The analysis considers a set of controlling factors that are known to be
significant drivers of low cloud behavior, which can be expressed as follows:

$$\text{CRE\_lcld}' \approx \sum_{i=1}^{7} \frac{\partial \text{CRE\_lcld}}{\partial X_i} \times X_i', \qquad (A2)$$


where CRE_lcld represents the cloud radiative effect from low-level clouds and the factors $(X_i)$
included in our analysis are 1) sea surface temperatures, 2) estimated inversion strength, 3)
horizontal surface temperature advection, 4) relative humidity at 700 hPa, 5) vertical velocity
at 700 hPa, and 6) near-surface wind speed. These parameters represent a combination of
thermodynamic and dynamic influences that are critical in dictating low cloud formation and
persistence (Scott et al., 2020). In addition to these standard meteorological variables, we
introduce 7) aerosol concentrations as additional controlling factors (Wall et al., 2022).
Specifically, we consider the natural logarithm of sulfate aerosol mass concentrations at 925
hPa, $\ln(SO_4)$. In our analysis, we opt to use data from the 925 hPa atmospheric level instead of
surface-level measurements. This decision is based on the understanding that conditions at 925
hPa provide a more accurate reflection of CCN concentrations near the cloud base (Painemal
et al., 2017). This altitude is often closer to the actual height at which low-level clouds form,
making it a more relevant indicator for assessing aerosol-cloud interactions. We also consider
the natural logarithm of the aerosol index, $\ln(AI)$ as a metric of the aerosol concentration cloud
controlling factor. Note that, as highlighted in the main text, since AI provides column-
integrated quantities and does not account for the vertical profile, it may not accurately capture
aerosol concentrations in low-level clouds, which are the focus of our study.

For each grid point, we employ ordinary least-squares multilinear regression to model
CRE_lcld' against anomalies in the seven cloud controlling factors. The regression coefficients,
$\partial \text{CRE\_lcld}/\partial \ln(SO_4)$ and $\partial \text{CRE\_lcld}/\partial \ln(AI)$, quantify the sensitivity of low-level cloud



radiative effect anomalies (CRE_lcld′) to local anomalies in ln(SO₄) or ln(AI), respectively.

**A4 Estimating ERFaci using CMIP6 model outputs**
**A4.1 Estimating ERFaci_true**
The ERFaci_true is calculated for PD minus PI conditions from aerosol-only, fixed-SST
experiments as,

$$ERFaci\_true = \Delta CRE\_lcld, \qquad (A3)$$

where the low-level cloud radiative effect ($\Delta CRE\_lcld$) is determined by using cloud
classification method introduced in Webb et al. (2006) and Soden and Vecchi (2011).

**A4.2 Estimating ERFaci_SC17**
This method partitions the low-level cloud radiative response observed in historical
experiments into two components: one is a temperature-mediated component (i.e., cloud
feedback) attributable to changes in the global-mean surface temperature and the other to
aerosol-cloud interactions. The temperature-mediated component is estimated by multiplying
the global-mean temperature anomaly by the low-level cloud feedback, derived from the
1pctCO₂ scenario ($\alpha_{1pctCO_2}$), which is calculated as the low-level cloud radiative response
normalized by the corresponding global-mean surface warming. This estimate of ERFaci is
then obtained by subtracting this temperature-driven component from the low-level cloud
radiative response, thus focusing solely on the impact of aerosol-cloud interactions.

$$ERFaci_{SC17} = \Delta CRE\_lcld - \alpha_{1pctCO_2} \cdot \Delta \overline{T}_s. \qquad (A4)$$

Because this method uses outputs from historical and 1pctCO₂ simulations, it allows a much
larger sample size of models to evaluate the two different versions of ERFaci_est.

**A4.3 Estimating ERFaci_est**
To estimate ERFaci_est, derived exclusively from CMIP6 model outputs calculated using
equations (1) and (2) from the main text, we use monthly anomalies spanning from 2000 to
2014 in historical experiments for susceptibility calculation, after removing trends and
climatological seasonality. We adhere to the same timeframe for aerosol concentration changes



as described in the main text. Additionally, given the challenges associated with deriving cloud-
top cloud droplet number concentrations ($N_d$) directly from CMIP6 model outputs, we adopt
an alternative approach, which is the maximum $N_d$ within a vertical atmospheric column
(Saponaro et al., 2020; Jia and Quaas, 2023). Owing to the limited availability of models for
CCF analysis and LWP binning, both are not explicitly employed in the estimation process of
ERFaci_est. Instead, we assess the impact of including or excluding CCF analysis and LWP
binning on ERFaci_obs to elucidate their influence on the estimation of ERFaci_est. The
simplified version of equations (1) and (2), which do not account for CCF analysis and LWP
binning, are presented below:

$$\text{ERFaci\_obs} \approx \frac{\partial \text{CRE\_lcld}}{\partial \ln(Y)} \times \Delta \ln(Y), \qquad (A5)$$

(without CCF analysis, LWP binning, and activation rate)

$$\text{ERFaci\_obs} \approx \left( \frac{\partial \text{CRE\_lcld}}{\partial \ln(N_d)} \times \frac{\partial \ln(N_d)}{\partial \ln(Y)} \right) \times \Delta \ln(Y), \qquad (A6)$$

(without CCF analysis and LWP binning but with activation rate)

When applying these equations to estimate ERFaci_obs, we obtain best estimates of global-
mean ERFaci_obs (without activation rate) of -1.46 for $SO_4$ and -1.74 for AI, and global-mean
ERFaci_obs (with activation rate) of -0.61 for $SO_4$ and -0.34 for AI. These values are 1.85,
1.53, 1.56, and 1.42 times larger, respectively, than those obtained when considering CCF
analysis and LWP binning. In other words, by dividing model-driven ERFaci estimates by these
factors, we can approximate its value under scenarios that include CCF analysis and LWP
binning (ERFaci_est). These outcomes are employed in Fig. 3 and Fig. A3.

**A5 Radiative Kernel Method**
Originally developed by Soden et al. (2008) to facilitate the analysis of radiative feedbacks,
"radiative kernels" describe the differential response of radiative fluxes to incremental changes
in the radiative state variables (e.g., clouds, temperature, water vapor, albedo). In this study, we
employed radiative kernel techniques derived from the HadGEM3-GA7.1 model (Smith et al.,
2020b) for all CMIP6 model analysis to isolate the genuine cloud radiative effect without
interference from cloud masking effects.

**A6 Estimating Global-Mean ERFaci_obs**





Given that our observation data cover the domain extending from 50°S to 50°N over the ocean, it is imperative to extrapolate global ERFaci values for comparison with the observation-based estimates reported in the IPCC Sixth Assessment Report. Our estimate of the ERFaci_obs spans a near-global domain, encompassing almost 60% of the Earth's surface. This notably includes vast stretches of the remote oceans. Although our estimate does not account for polar oceans, their exclusion is unlikely to significantly skew our results. These regions contribute minimally to the global ERFaci because of their limited surface area. Given these considerations, we believe that our near-global estimate can serve as a reliable proxy for the true global average. This assumption is supported by the result from CMIP6 models (Fig. A4). To bridge the gap between global and domain-specific averages, using 21 CMIP6 climate models in single-forcing experiments (ERFaci_true), we employ a scalar, $\gamma$, representing the ratio of the multi-model mean of global-average ERFaci_true to the multi-model mean of domain-average ERFaci_true. We ascertain $\gamma$'s value at 0.69 with 0.92 correlation coefficient, enabling the calibration of our domain-specific ERFaci estimates to more accurately reflect a global scale. This calibration is achieved through the following equation:

$$\text{ERFaci\_obs, global} = \gamma \times \text{ERFaci\_obs, domain}, \qquad (A7)$$

In ensuring the consistency of our estimates, we adjust the IPCC Sixth Assessment Report's estimate of ERFaci, which uses 2014 as the present-day reference year and 1750 as the preindustrial reference year. The IPCC's initial global estimate for ERFaci between 2014 and 1750 is $-1.0 \pm 0.7$ W m$^{-2}$. To make this preindustrial reference period consistent with our analysis, we subtract the estimated ERFaci of $-0.07$ W m$^{-2}$ between 1850 and 1750 from the IPCC's value (Dentener et al., 2021). This adjustment yields an estimate based solely on observational evidence, with a 90% CI of $-0.93 \pm 0.7$ W m$^{-2}$ (Wall et al., 2022).

**A7 Uncertainty**

The uncertainty in ERFaci_obs, in the case where the activation rate is not considered, is attributed to uncertainties in the susceptibility, the regression coefficient for $\partial\text{CRE\_lcld}/\partial\ln(Y)$, and in the model estimates of $\Delta\ln(Y)$. Conversely, when considering the activation rate, the uncertainty in ERFaci_obs stems from uncertainties in the regression coefficients for $\partial\text{CRE\_lcld}/\partial\ln(N_d)$ and $\partial\ln(N_d)/\partial\ln(Y)$, as well as from uncertainties in the model predictions of $\Delta\ln(Y)$.






To quantify the uncertainty derived from regression coefficients, at each grid box a 90%
confidence interval of the susceptibility is given by

$$\delta = t\sqrt{\mathbf{C}_{ii}}\sqrt{\frac{N_{nom}}{N_{eff}}} \text{ (without activation rate),} \qquad (A8)$$

$$\delta = t\sqrt{\Delta x^T \mathbf{C} \Delta x}\sqrt{\frac{N_{nom}}{N_{eff}}} \text{ (with activation rate),} \qquad (A9)$$

where t is the critical value of the Student's t-test at the 95% significance level with $N_{eff} - 7$
degrees of freedom (Von Storch and Zwiers, 1999), $\mathbf{C}$ is the variance–covariance matrix of
regression coefficients hence $\mathbf{C}_{ii}$ represents the diagonal components of the $\mathbf{C}$, $N_{nom}/N_{eff}$ is
the ratio of the nominal to effective number of monthly values of CRE_lcld′, and $\Delta x$ is the
regression coefficient for $\partial \ln(N_d)/\partial \ln(Y)$. $\mathbf{C}$ is formulated as $\mathbf{C} = \hat{\sigma}^2(X^TX)^{-1}$, where X is
the data matrix with columns composed of detrended monthly anomalies. Specifically, these
anomalies are of $\ln(Y)$ in scenarios where the activation rate is not considered and of $\ln(N_d)$
in scenarios where the activation rate is included. The term $\hat{\sigma}^2$ denotes the mean of squared
residuals of the regression model and we estimate $N_{nom}/N_{eff}$ as $(1 + r)/(1 - r)$, where r is
the lag one autocorrelation of CRE_lcld′.

Uncertainty for spatially averaged regression coefficients is calculated as

$$\Delta_{obs} = \sqrt{\frac{\sum_{k=1}^{N_{nom}^*}(\delta_k w_k)^2}{\left(\sum_{k=1}^{N_{nom}^*} w_k\right)^2}}\sqrt{\frac{N_{nom}^*}{N_{eff}^*}} \ , \qquad (A10)$$

where $\delta_k$ denotes the uncertainty of the $k^{th}$ grid box, $w_k$ is the cosine of the latitude. $N_{nom}^*$
represents the nominal number of spatial degrees of freedom, while $N_{eff}^*$ represents the
effective number of spatial degrees of freedom. The ratio $N_{nom}^*/N_{eff}^*$ is determined through
empirical orthogonal function (EOF) analysis applied to CRE_lcld' for all ocean grid boxes
between 50°S and 50°N as outlined in equation 5 of Bretherton et al. (1999). Before conducting
the EOF analysis, each grid of CRE_lcld' value is multiplied by $\sqrt{w_k}$ to mitigate dependencies
on grid geometry (North et al. 1982). The derived value of $\Delta_{obs}$ quantifies the half-width of the
90% CI for ERFaci_obs over our domain region specifically reflecting the uncertainty
associated with regression coefficients.



To estimate uncertainty derived from model predictions, we examine the entire range of aerosol
concentration changes across each CMIP6 model, instead of estimating uncertainty within the
$5^{th}$-$95^{th}$ percentile range, primarily due to the limited number of models available for our
analysis: 13 models for $\Delta \ln(SO_4)$ and 9 models for $\Delta \ln(AI)$. This decision reflects a
methodological adaptation to the limited model dataset, ensuring a comprehensive evaluation
of model-derived uncertainty (Myers et al., 2021). We first calculate ERFaci_obs by
multiplying $\Delta \ln(Y)$ from each of the models by the observationally derived susceptibility. The
half-width of the CI, denoted as $\Delta_{model}$, is derived by halving the difference between the
maximum and minimum estimates of ERFaci_obs. The overall 90% CI is determined by
$\text{ERFaci\_obs, domain} \pm \sqrt{\Delta_{obs}^2 + \Delta_{model}^2}$.

In our methodology, the scalar $\gamma$ is used to extrapolate the global ERFaci_obs from our domain-
specific ERFaci_obs estimates. This extrapolation introduces an additional component of
uncertainty. Although both $\gamma$ and the changes in aerosol concentration are obtained from
CMIP6 model outputs, it's important to note that $\gamma$ does not directly correlate with aerosol
concentration changes across the models. Consequently, the uncertainty associated with $\gamma$ is
quantified using the root mean squared error (RMSE) between the domain-specific averaged
ERFaci_true, multiplied by $\gamma$, and the global-mean ERFaci_true. The overall 90% CI is
determined by $\text{ERFaci\_obs, global} \pm \sqrt{([\gamma]\Delta_{obs})^2 + ([\gamma]\Delta_{model})^2 + \Delta_{\gamma}^2}$, where square
brackets indicate multi-model mean of a parameter.


**Author Contributions:** B.J.S. designed research; C.P. performed research; C.P. analyzed
data; B.J.S., R.J.K., T.S.L., and H.H. contributed ideas; C.P., B.J.S., R.J.K., T.S.L., and H.H.
wrote the paper.

**Competing Interest Statement**: The authors declare no competing interest.

**Acknowledgements**
We greatly wish to thank Casey J. Wall for sharing the post-processed CERES and MERRA-
2 reanalysis data used in Wall et al. (2022), and Edward Gryspeerdt for sharing data related to



cloud droplet number concentration. C.P. and B.J.S. were supported by the National Oceanic
and Atmospheric Administration Climate Program Office's Modeling, Analysis, Predictions,
and Projections Program Grant NA21OAR4310351. R.J.K. was supported by NASA Science
of Terra, Aqua and Suomi-NPP grant no. 80NSSC21K1968. T.S.L. was supported by National
Aeronautics and Space Administration CloudSat Grant G-39690-1.

## Data Availability

CERES data were downloaded from the National Aeronautics and Space Administration
(NASA) CERES ordering tool (https://ceres.larc.nasa.gov/data/). MODIS data were
downloaded from NASA Level-1 and Atmosphere Archive and Distribution System
(https://ladsweb.modaps.eosdis.nasa.gov/archive/allData). MODIS $N_d$ data are available from
the Centre for Environmental Data Analysis
(https://doi.org/10.5285/864a46cc65054008857ee5bb772a2a2b, Gryspeerdt et al., 2022).
MERRA-2 reanalysis data were downloaded from NASA Goddard Earth Sciences Data and
Information Services Center (https://doi.org/10.5067/LTVB4GPCOTK2). The CMIP6 data
used in this study are available at the Earth System Grid Federation data portal (https://esgf-
node.llnl.gov/projects/cmip6/). Intermediate data products used in our analysis, including
gridded monthly anomalies and regression coefficients, are available from GitHub
(https://github.com/nicklutsko/Radiative_Forcing_Aerosol_Clouds, Wall et al., 2022).

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

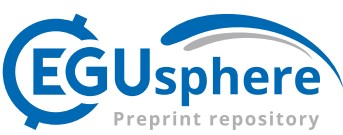

**Figures**

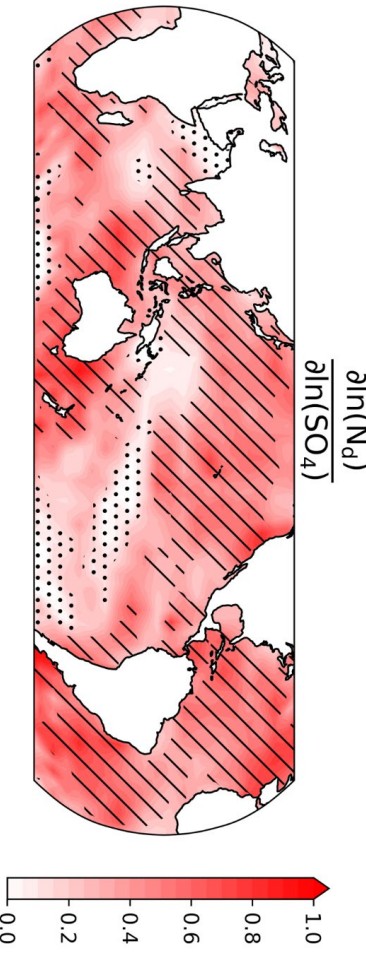

**Fig. 1.** Regression coefficient map of the activation rate of cloud droplet number concentration ($N_d$) to sulfate aerosol concentration ($SO_4$). The color scale indicates the magnitude of sensitivity, where an increase in $SO_4$ concentration corresponds to an increase in $N_d$. Areas with diagonal indicate correlation coefficients exceeding 0.4, demonstrating a significantly high linearity between $SO_4$ and $N_d$. Areas with stippling indicate where the changes are not statistically different from zero at the 95% confidence level using a Stduent's t-test.





**Fig. 2.** Spatial distribution of ERFaci_obs components and the estimated ERFaci_obs differentiated by the consideration of



the activation rate. (a) Multi-model mean (MMM) of changes in $SO_4$ concentration between pre-industrial (PI) and present-
day (PD) periods. 13 models are used for this analysis (Table A1). (b,c) Susceptibility of low cloud radiative effect to $SO_4$
concentration derived from CCF analysis using observations (Appendix A). (d,e) Observationally constrained ERFaci for
$SO_4$ estimated by multiplying the susceptibility with the changes in $SO_4$ concentration.







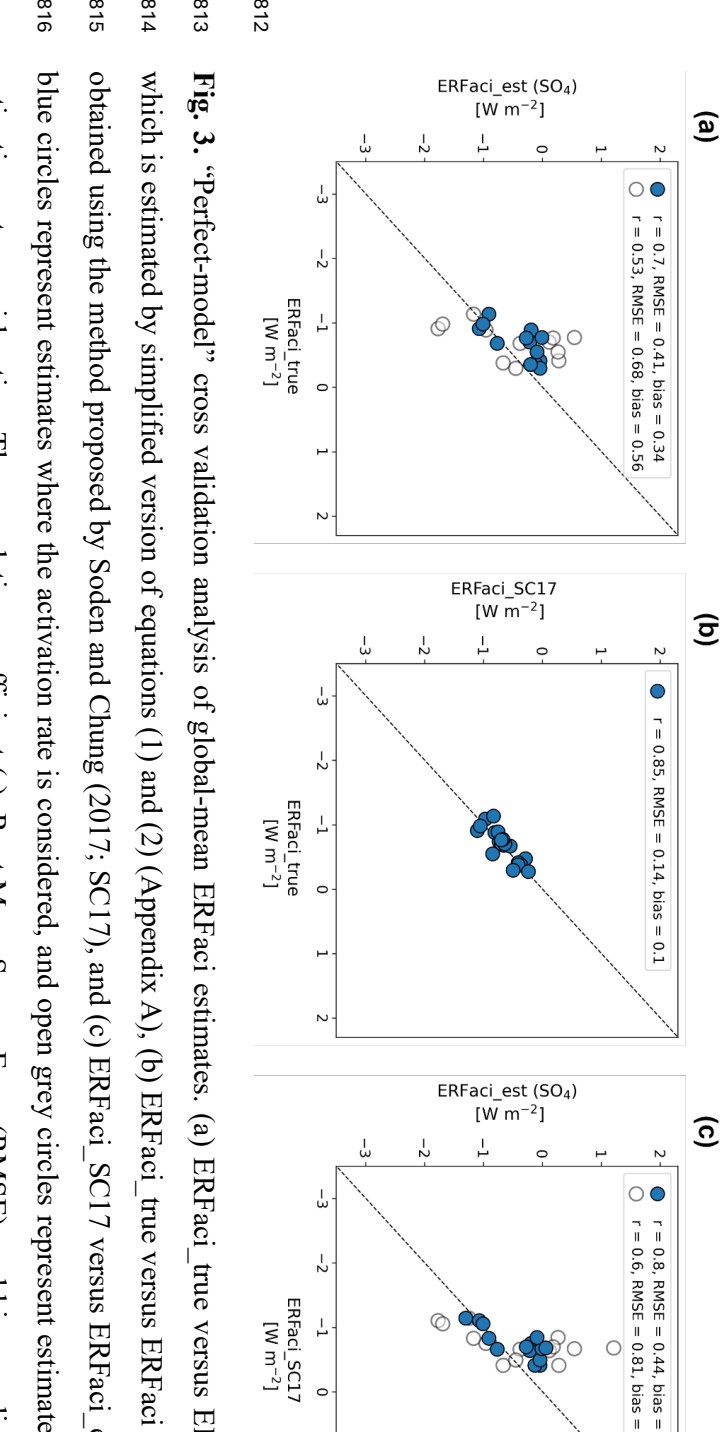

**Fig. 3.** "Perfect-model" cross validation analysis of global-mean ERFaci estimates. (a) ERFaci_true versus ERFaci_est which is estimated by simplified version of equations (1) and (2) (Appendix A), (b) ERFaci_true versus ERFaci estimates obtained using the method proposed by Soden and Chung (2017; SC17), and (c) ERFaci_SC17 versus ERFaci_est. Filled blue circles represent estimates where the activation rate is considered, and open grey circles represent estimates without activation rate consideration. The correlation coefficient (r), Root Mean Square Error (RMSE), and bias are displayed in the upper left corner of each panel. Bias is defined as the mean absolute difference from the 1:1 reference line, depicted by a dashed line. All panels have identical x and y axis ranges to highlight the variance among the estimation methods. Higher r values, lower RMSE, and minimal bias indicate consistency in ERFaci estimates across different estimation methods using CMIP6 models.



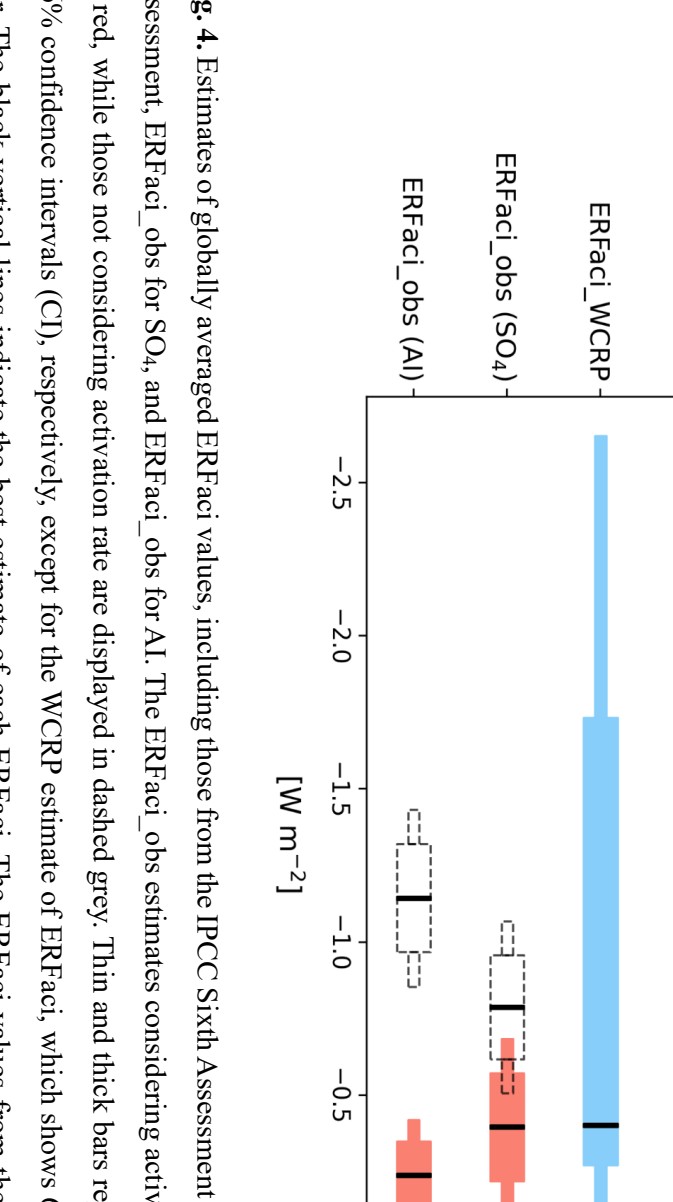

**Fig. 4.** Estimates of globally averaged ERFaci values, including those from the IPCC Sixth Assessment Report, from WCRP assessment, ERFaci_obs for SO$_4$, and ERFaci_obs for AI. The ERFaci_obs estimates considering activation rate are shown in red, while those not considering activation rate are displayed in dashed grey. Thin and thick bars represent the 90% and 66% confidence intervals (CI), respectively, except for the WCRP estimate of ERFaci, which shows 68% CI for the thick bar. The black vertical lines indicate the best estimate of each ERFaci. The ERFaci values from the IPCC represent the assessment based on observational evidence alone.



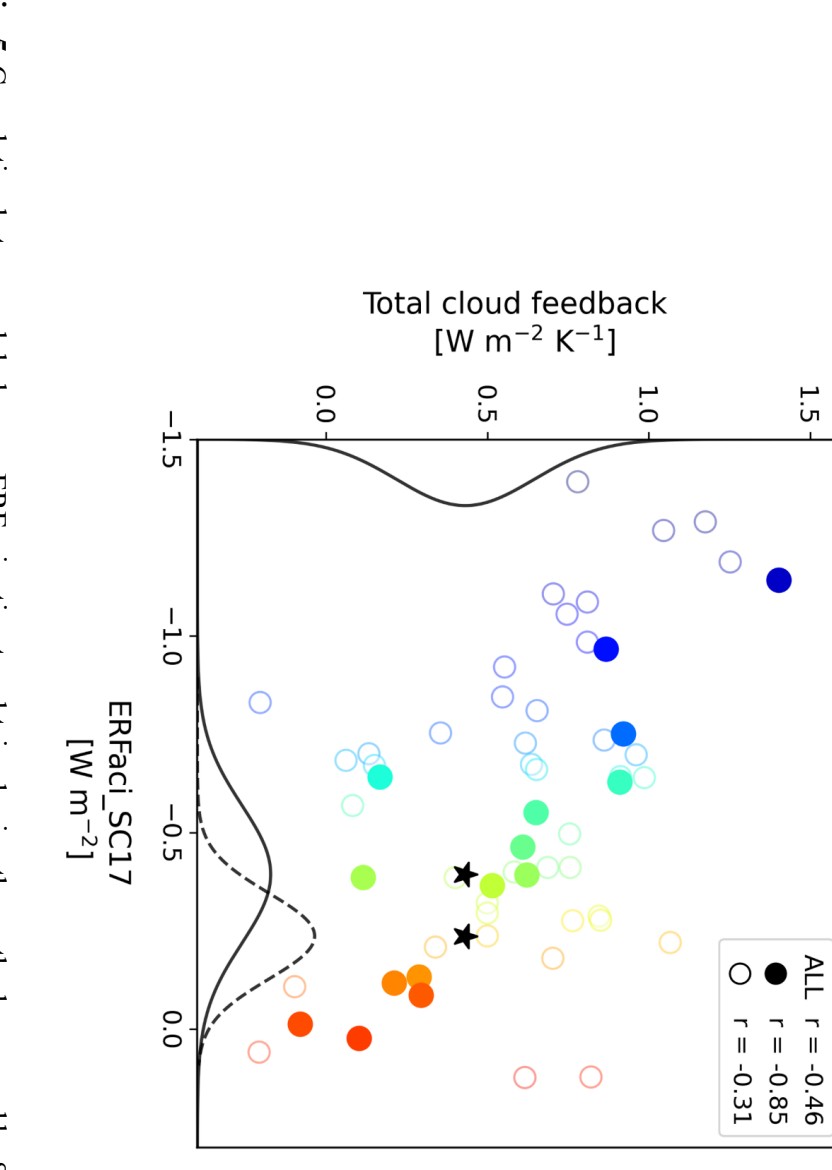

**Fig. 5.** Correlation between global-mean ERFaci estimates obtained using the method proposed by Soden and Chung (2017;

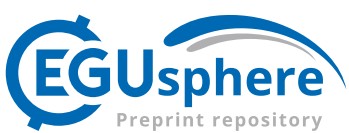

SC17), aimed at expanding the model availability, and the globally averaged total cloud feedback as determined by the
corresponding models. Each dot represents a single model. The colors from red to blue indicate weak ERFaci models to
strong negative ERFaci models. Filled circles represent the 15 'GOOD HIST' models that align more closely with historical
observations of global-mean surface warming, whereas open circles denote the remaining models (Appendix A).
Correlation coefficients (r) for the entire models, the 'GOOD HIST' models, and remaining models are shown in the upper
right corner. The probability density functions (PDFs) showing the 90% confidence intervals for observationally constrained
ERFaci from sulfate concentration (SO₄; solid line) and the aerosol index (AI; dashed line) are plotted along the x-axis,
while the PDF for observationally constrained total cloud feedback (solid line), derived from Ceppi and Nowack (2021), is
plotted on the y-axis (amplitudes scaled arbitrarily). Stars denote the best estimates of the PDFs, signifying the most
probable values within the distributions.











**Fig. A1.** Same as Fig. 1 but for AI.

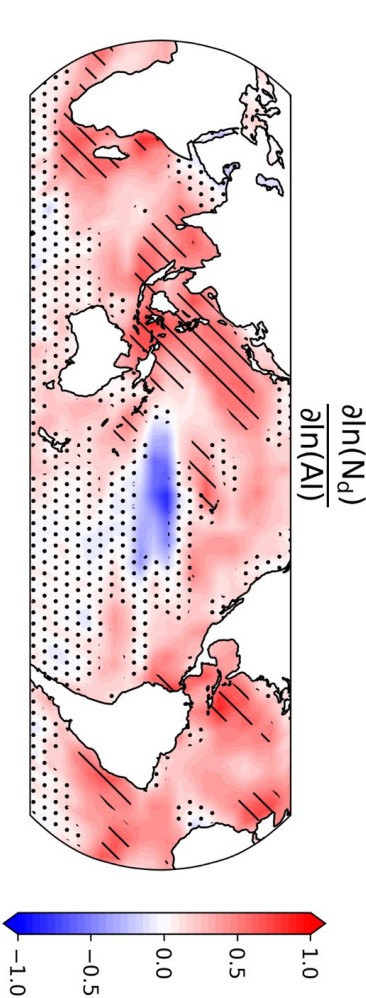



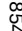








**Fig. A2.** Same as Fig. 2 but for AI. 9 models are used for changes in AI (Table A1).









**Fig. A3.** Same as the first and last scatter plots in Fig. 3 but for the ERFaci_est estimated by AI instead of SO₄.

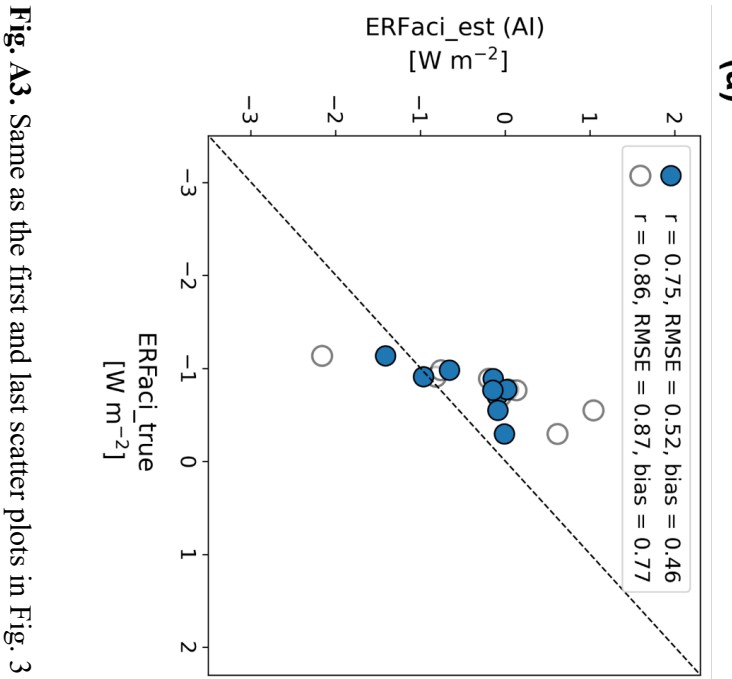



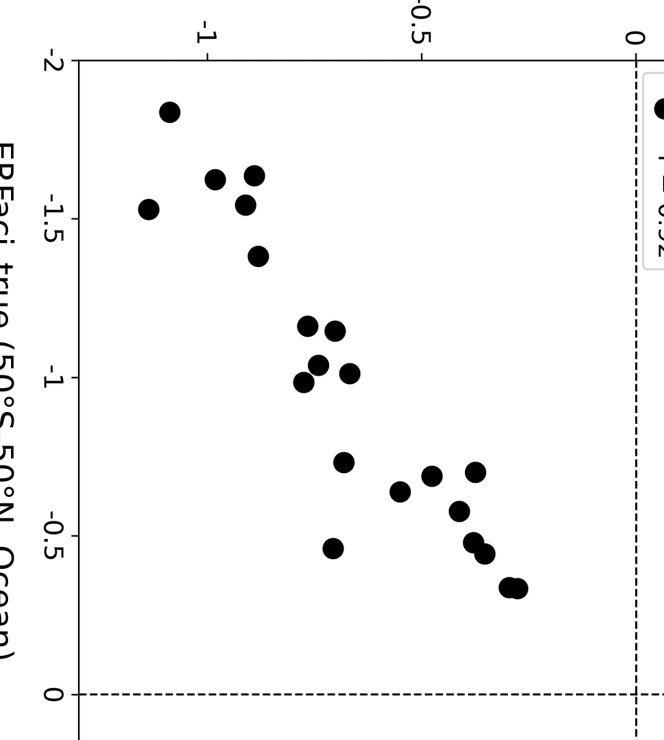

**Fig. A4.** CMIP6 estimates of ERFaci_true, averaged for the domain region (50°S to 50°N over ocean), and globally averaged ERFaci_true values. Each black circle represents an individual model's estimate, with the correlation coefficient (r) indicated in the upper left corner.



869 **Table A1**. CMIP6 models used in the analysis.

| | Model | Δln(SO₄) | Δln(AI) | ERFaci_true | ERFaci_SC17 | ERFaci_est (SO₄) | ERFaci_est (AI) | GOOD HIST index |
|---|---|---|---|---|---|---|---|---|
| 1 | ACCESS-CM2 | | | o | o | | | 0.323 |
| 2 | ACCESS-ESM1-5 | | | o | o | | | 0.184 |
| 3 | AWI-CM-1-1-MR | | | | o | | | 0.074 |
| 4 | AWI-ESM-1-1-LR | | | | o | | | 0.141 |
| 5 | BCC-CSM2-MR | | | | o | | | 0.319 |
| 6 | BCC-ESM1 | o | | o | o | o | | 0.448 |
| 7 | CAMS-CSM1-0 | | | | o | | | 0.268 |
| 8 | CanESM5 | | | o | o | | | 0.169 |
| 9 | CanESM5-1 | | | | o | | | 0.248 |
| 10 | CanESM5-CanOE | | | | o | | | 0.306 |
| 11 | CAS-ESM2-0 | | | | o | | | 0.366 |
| 12 | CESM2 | | | o | o | | | 0.147 |
| 13 | CESM2-FV2 | | | | o | | | 0.288 |
| 14 | CESM2-WACCM | | | | o | o | | 0.104 |
| 15 | CESM2-WACCM-FV2 | | | | o | | | 0.372 |
| 16 | CIESM | | | | o | | | 0.212 |
| 17 | CMCC-CM2-SR5 | | | | o | | | 0.173 |
| 18 | CMCC-ESM2 | | | | o | | | 0.165 |
| 19 | CNRM-CM6-1 | | | o | o | | | 0.029 |
| 20 | CNRM-CM6-1-HR | | | | o | | | 0.014 |
| 21 | CNRM-ESM2-1 | o | | o | o | o | o | 0.191 |
| 22 | E3SM-1-0 | | | | o | | | 0.289 |
| 23 | E3SM-2-0 | | | | o | | | 0.749 |
| 24 | EC-Earth3 | | | o | o | | | 0.136 |
| 25 | EC-Earth3-AerChem | o | o | o | o | o | o | 0.362 |
| 26 | EC-Earth3-CC | | | | o | | | 0.503 |
| 27 | EC-Earth3-Veg | | | | o | | | 0.153 |
| 28 | EC-Earth3-Veg-LR | | | | o | | | 0.127 |
| 29 | FGOALS-f3-L | | | | o | | | 0.115 |
| 30 | FIO-ESM-2-0 | | | | o | | | 0.256 |
| 31 | GFDL-CM4 | o | | o | o | o | | 0.242 |
| 32 | GFDL-ESM4 | o | o | o | o | o | o | 0.43 |
| 33 | GISS-E2-1-G | | | o | o | | | 0.347 |
| 34 | GISS-E2-1-H | | | | o | | | 0.115 |
| 35 | GISS-E2-2-G | | | | o | | | 0.272 |
| 36 | GISS-E2-2-H | | | | o | | | 0.115 |
| 37 | HadGEM3-GC31-LL | o | o | o | o | o | o | 0.191 |
| 38 | HadGEM3-GC31-MM | | | | o | | | 0.284 |
| 39 | ICON-ESM-LR | | | | o | | | 0.287 |
| 40 | INM-CM4-8 | | | | o | | | 0.134 |
| 41 | INM-CM5-0 | | | | o | | | 0.201 |
| 42 | IPSL-CM5A2-INCA | | | | o | | | 0.293 |
| 43 | IPSL-CM6A-LR | | | o | o | | o | 0.157 |
| 44 | IPSL-CM6A-LR-INCA | o | | o | | | | 0.081 |
| 45 | KACE-1-0-G | | | | o | | | 0.147 |
| 46 | KIOST-ESM | | | | o | | | 0.15 |
| 47 | MIROC6 | o | o | o | o | o | o | 0.327 |
| 48 | MIROC-ES2L | | | | o | o | | 0.296 |
| 49 | MPI-ESM1-2-HR | | | | o | | | 0.15 |
| 50 | MPI-ESM1-2-LR | | | | o | | | 0.072 |
| 51 | MPI-ESM-1-2-HAM | o | o | o | o | o | o | 0.507 |
| 52 | MRI-ESM2-0 | o | o | o | o | o | o | 0.329 |
| 53 | NESM3 | | | | o | | | 0.216 |
| 54 | NorCPM1 | | | | o | | | 0.17 |
| 55 | NorESM2-LM | o | o | o | o | o | o | 0.455 |
| 56 | NorESM2-MM | o | o | o | o | o | o | 0.366 |
| 57 | SAM0-UNICON | | | | o | | | 0.362 |
| 58 | TaiESM1 | | | | o | | | 0.417 |
| 59 | UKESM1-0-LL | o | o | o | o | o | o | 0.325 |
| 60 | UKESM1-1-LL | | | | o | | | 0.098 |

870