# Peer review of "Observational Constraints Suggest a Smaller Effective Radiative Forcing from Aerosol-Cloud Interactions"

_EGUsphere, 2024_

## Author Comment (AC1)

**The response to Reviewer #1:**

The article "Observational Constraints Suggest a Smaller Effective Radiative Forcing from Aerosol-Cloud Interactions" by Park et al. quantifies the effective radiative forcing from aerosol-cloud interactions to better represent climate predictions. The authors used satellite observations, reanalysis and climate models to support their analysis and results, and they focus on the effect of aerosols on cloud droplet concentration and evaluate the role of aerosol activation, which is usually overlooked in current estimates of effective radiative forcing.

I think this topic is within the scope of the ACP, as it seems to be an important parameter that the scientific community should take into account. The hypothesis is well explained and we understand each step of the process, but there are parts that should be improved before publication on ACP. I think the main problem is the predominance of the appendix and important information should be included in the main article. I am also concerned about the omission of meteorological parameters in the study and I do not know how this affects the study of cloud controlling factors. I recommend the paper for publication after the major revisions I suggest below.

We sincerely thank the reviewer for their constructive comments and valuable suggestions. As our manuscript has been submitted to *ACP Letters*, which has strict word limits, we have prioritized presenting the main findings concisely. Where necessary, additional details have been included in the Appendix to provide further support for our findings. We kindly ask for your understanding in this regard. Below, we provide specific responses to each comment, highlighted in blue. All "L" references in our responses correspond to line numbers in the revised manuscript with highlighted changes.

**Major Comments**

1. The conclusion does not follow the ACP recommendations. The conclusion needs to be expanded.

"*Every article must have a final section where the overall advances are concisely summarized and put in context. Although the results section may include some discussion, a synthesis and interpretation must appear in the final section. ACP expects that the concluding section will normally include the following components, although not necessarily in separate paragraphs:*

*\* Summary: Summarize the main results and relate them to the objectives, questions, or hypotheses of the study. The summary should include the main quantitative results.*

*\* Synthesis/interpretation: Explain and interpret the results concisely to enable readers to make sense of them as a whole.*

*\* Comparison and context: Compare the results with previous studies to put them in context. Explain consistencies, inconsistencies, and advances in knowledge.*

*\* Caveats and limitations: State how these affect confidence in the overall results, and where*

*future work is needed.*

*\* Implications: Discuss what the results mean for our understanding of the state and/or behaviour of the atmosphere and climate, which is the main requirement for publication in ACP. The editor's acceptance/rejection decision will be strongly guided by this component of the concluding section.*"

Thank you for your valuable feedback. In response, we expand the conclusion section to align more closely with the journal's recommendations. This revision now provides a comprehensive discussion of the uncertainties our study has, ensuring a balanced view of the findings. However, given the strict word limit of *ACP Letters*, it is challenging to provide more details. We appreciate your understanding regarding these constraints and thank you for your thoughtful review.

2. Aerosols have an effect on cloud properties, cloud droplet size and droplet concentration, but the effect is small compared to the effect of meteorological parameters. If the authors did not constrained the results of Fig. 1 by meteorological parameters, then the changes in Nd can be due to meteorological parameters and what is observed is (indirectly) the correlation between aerosols and meteorological parameters. If I understand correctly, these coefficients are used afterward in the Equation A6. This problem is taken into account for the cloud controlling factors but I am not sure about the impact for the study.

Thank you for your comment and suggestion. In the revised manuscript, we now account for environmental factors when assessing the activation rate of aerosols into cloud droplets. This ensures that meteorological influences do not confound the activation rate analysis, as reflected in L88-89, where we state: "To verify the key assumption while accounting for environmental influences, we performed cloud controlling factor (CCF) analysis (Appendix A3).". This approach results in consistent findings, supporting the robustness of our conclusions. Additionally, in the revised manuscript, we extend this consideration to include not only activation rate metrics but also the susceptibility and the resulting effective radiative forcing due to aerosol-cloud interactions (ERFaci) estimates, particularly for those equations explicitly incorporating activation rate.

3. Section A3 : Cloud controlling factor analysis,

Have the authors attempted to perform a Variance Inflation Factors analysis to estimate the performance of the CCF (as done in Scott et al. 2020) to avoid any cross-correlation and ensure that the effect seen is due to aerosols only?

Thank you for your comment. In response, we perform a Variance Inflation Factor (VIF) analysis, following the methodology in Scott et al. (2020), to evaluate any potential multicollinearity within CCF analysis. The results indicate that the sulfate mass concentrations ($SO_4$) and aerosol index (AI) are independent of other environmental factors, supporting the validity of our approach and emphasizing that our estimates of ERFaci are truly attributable to aerosols. We include these VIF results in Figures A1 and A2, along with corresponding explanations in Appendix A3.

4. L457: The authors state that their value is the same as the global value because the polar ocean surfaces are limited in area. Firstly, we can argue whether the region is indeed negligible in terms of area compared to the globe, but their impact could be significantly greater in these regions due to their specificities (polar night/day, ice surface, pristine conditions…). I am currently not convinced that the results can really be generalised to the globe.

Thank you for your comment. As noted in the manuscript, limitations in satellite observations prevent us from obtaining reliable data over land and polar regions (Jia et al., 2019; Gryspeerdt et al., 2022; Jia and Quaas, 2023). However, we expand our study domain to cover the area between 60°S and 60°N, and our conclusions remain consistent within this extended area. We believe that excluding the polar regions does not compromise our conclusions, as sulfate mass concentrations are primarily concentrated in major industrial regions, such as East Asia and North America, which are well represented within our study domain.

Additionally, to address your concern, we adjust the text to avoid potential confusion regarding polar region contributions. We now highlight our extrapolation approach using CMIP6 single-forcing experiments, where we apply a scalar multiplier to the domain-average value to estimate a global-average value, as explained in Appendix A6 and illustrated in Figure A3.

5. A3, have the authors considered different cloud regimes as in Scott et al 2022?

Thank you for your comment. In this study, we focus specifically on non-obscured low clouds without distinguishing between cloud types, as our primary objective is to examine observationally constrained global effective radiative forcing from aerosol-cloud interactions and validate it through various estimation methods, based on CMIP6 models. Given the dominant contribution of low-level clouds to ERFaci, focusing on these cloud types is sufficient to represent the ERFaci value (Christensen et al., 2016; Bellouin et al., 2020; Forster et al., 2021). While considering different cloud regimes, as suggested by Scott et al. (2022), could provide valuable insights, the journal's word limit necessitated focusing on non-obscured low clouds to maintain clarity and conciseness. We agree that exploring cloud regime distinctions would be an excellent avenue for future research.

6. Statistical tests and quantification would be welcome to better assess the results instead of "significantly diminishes" for example, etc.

Thank you for your comment. We revise the manuscript to remove the term "significantly" where statistical quantifications are not applied. Additionally, we include $p$-values alongside $r$-values where correlation coefficients are reported for improved statistical rigor.

**Minor :**

The ACP guidelines mention : "*Appendices: all material required to understand the essential*

*aspects of the paper such as experimental methods, data, and interpretation should preferably be included in the main text.*" I have found that most ACP papers have data set and methods sections and usually an appendix with important information. I strongly recommend to include the data set and method sections in the main body of the paper and not in the appendix, following the ACP recommendations.

Thank you for your comment and recommendation. To address your concern, we specify the origin of each dataset in the main text for improved clarity. We considered moving the methods for estimating ERFaci in CMIP6 models to the main text in accordance with ACP guidelines; however, due to the strict word limit for *ACP Letters*, we decide to retain the current format. We appreciate your understanding.

L14: "While some studies it is assumed", I suggest "While some studies assumed"

Thank you for your suggestion. We revise the sentence to "While some studies assume…".

L15: "Variation in sulfate aerosols", do the authors mean sulphate aerosol concentration? It could also be aging, coating, etc that would change the aerosol-cloud interaction.

Thank you for your valuable comment. We were indeed referring to sulfate aerosol mass concentrations in the atmosphere. We revise the text to specify "sulfate aerosol mass concentrations" or "sulfate mass concentration" where relevant to avoid any potential confusion.

L15: I think Sulfates are SO42- and not SO4

Thank you for your comment. To clarify, we add a note in L66 stating, "sulfate mass concentration ($SO_4$; for simplicity, we omit its ionic form)…" to ensure readers understand the notation used throughout the text. Additionally, we note that Randles et al. (2017), the MERRA-2 reference paper, also uses sulfate as $SO_4$. Therefore, we believe this notation aligns with established conventions and will be clear to readers.

L20: It would be interesting to know how much, on average, the ERF is reduced and less uncertain. A quantification would increase the impact of the abstract.

Thank you for your comment. Given our use of two different aerosol concentration proxies, we instead add quantification of recent climate assessments. The revised sentence now states in L21 "Our results suggest a smaller and less uncertain value of the global ERFaci ($-0.32 \pm 0.21$ W m$^{-2}$ for $SO_4$, 90% confidence) than recent climate assessments (e.g. $-0.93 \pm 0.7$ W m$^{-2}$, 90% confidence)…".

L34: All aerosols do not act as CCN, some would act as INP, and some would not interact with clouds.

Thank you for your comment. You are correct that not all aerosols serve as CCN; some act as INP, while others may not interact with clouds at all. In this study, we address aerosol-cloud interactions in a broad context, primarily focusing on aerosols that contribute to CCN. Additionally, this study focuses specifically on aerosol-cloud interactions within low-level clouds, where ice formation is minimal, and CCN-related interactions are most prominent and well-documented (Christensen et al., 2016; Bellouin et al., 2020; Forster et al., 2021). This generalization is intended to streamline the discussion and maintain relevance to our specific focus on CCN-driven interactions.

L47-51: Citations are missing to support the text.

Thank you for your comment. We revise the sentence and add references to support the text in L52-55: "In some studies, the activation rate is not explicitly incorporated into the estimation process of ERFaci as it is implicitly assumed to have a one-to-one relationship (e.g. Chen et al., 2014; Christensen et al., 2016; Douglas and L'Ecuyer 2020; Wall et al., 2022, 2023)."

L62: Have the authors constrained to consider situations where sulphate aerosols are the dominant aerosol type (e.g. more than 80% of the total concentration?) Other aerosols may not be as efficient CCN but they could still bias their results.

Thank you for your comment. The original sentence may have been misleading. Our intent was to convey that sulfate mass concentration is recognized as a primary contributor to cloud droplet formation among various aerosol types (Charlson et al., 1992; McCoy et al., 2018). To enhance clarity, we revise the sentence to state in L69-71: "$SO_4$ is recognized as a dominant contributor to cloud droplet formation, alongside other aerosol types such as black carbon, organic carbon, sea salt, and dust (Charlson et al., 1992; McCoy et al., 2018).".

L77: I do not think Nd is defined in the paper.

Thank you for your comment. We now define $N_d$ in the introduction section at L51 to ensure clarity.

L81: I would not expect a one-to-one relation. As the authors mentioned it is related to the activation rate but this would mean that SO4 is the only CCN.

Thank you for pointing this out. Of course, our intention is not to suggest that $SO_4$ is the only contributor to CCN, but rather to illustrate that not all sulfate aerosols convert to cloud droplets. Additionally, we stated in L69-71 that "$SO_4$ is recognized as a dominant contributor to cloud droplet formation, alongside other aerosol types…", which we believe clarifies that $SO_4$ is not the only CCN. However, to avoid any confusion, we revise the text to replace "one-to-one relation" with the sentence in L92: "…underscoring that not all $SO_4$ in the atmosphere are converted into cloud droplets.". Furthermore, we add in L98: "This variation may be attributed to differences in local environmental conditions and the role of aerosols in which these clouds

occur (e.g. Douglas and L'Ecuyer, 2019, 2020).". These revisions aim to ensure clarity and prevent misunderstanding.

L82-84 : "show a notably weaker (…) coefficient", bur the correlation coefficient, therefore shall we conclude anything from that ?

Thank you for pointing this out. To avoid confusion and improve clarity, we remove the correlation coefficient from Figure 1 and Figure S1, as it was not ideal for visually representing the activation rate.

Fig1: I am not sure what is the date range used by the authors to produce the plot.
Thank you for your comment. We add the date range (January 2003 to December 2019) to the figure caption for clarification.

L85: I do not find the results consistent between AI and SO4, there are negative values and the results with large regression coefficients are not the same. Can the authors clarify what they mean by consistent?

Thank you for pointing this out. You are correct that there are differences between the AI and SO$_4$ results. While both exhibit similar regional patterns—such as higher activation rates in stratocumulus regions, consistent with SO$_4$ findings—the results for $\partial\ln(N_d)/\partial\ln(AI)$ differ in some aspects. To clarify, we now state in L100-101: "Repeating our analysis using AI yields somewhat different results with those for SO$_4$ though still showing strong positive regression coefficients near continental coasts (Fig. S1).". Additionally, as noted in the manuscript, the differences in regression coefficients may arise from the use of column-integrated quantities (AI from MODIS), which do not capture the aerosol vertical structure. This clarification would help provide further context for the observed discrepancies.

L201 : "lower end", Do they authors mean "higher end" ?

Thank you for pointing this out. We revise the text to "higher end" and add "(less negative)" for additional clarification.

Equation A1: I am not sure to understand the equation, Ln is at the 2.5 horizontal resolution but L and U are at the native resolution but within the 2.5 degree grid cell, is it correct ?

Thank you for pointing this out. To improve clarity, we remove the expression "relative to the area of each grid box," as it causes confusion regarding the resolution of $L_n$, L and U in Equation A1. All variables are evaluated at the same 2.5° grid point resolution.

Equations A5 and A6, what is needed to account for LWP bins? If I understand, the authors have constrained for LWP, is 30 g cm-2 bins sufficient? Have they tried smaller bins to see how the results change?

Thank you for your comment. Initially, we considered using liquid water path (LWP) bins to further specify cloud properties based on LWP. However, to maintain consistency and directly compare our results with those of Wall et al. (2022), which do not constrain LWP, we decide to exclude LWP in our equations. Even without this constraint, our results remain consistent, supporting the robustness of our findings.

L558 and L561, I cannot access the webpages:

https://esgf559node.llnl.gov/projects/cmip6/ and
https://github.com/nicklutsko/Radiative_Forcing_Aerosol_Clouds

Thank you for your comment. I check the web addresses and, in place of the data from Wall et al. (2022) as we extend our domain area to 60°S and 60°N, I create a data repository for the relevant variables used in our study. The data are now accessible at https://zenodo.org/records/14058556.

I think all the appendices are not referenced in the main text. For example, I cannot see where sections A3 and A4 are referenced in the main article. It just says "In Appendix A".  Again I think most of the part should be in the main text but for the remaining part, I suggest to clearly specify which part is referenced in the main text.

Thank you for pointing this out. We revise the manuscript to specify exactly which appendix sections are referenced in the main text to improve clarity.

**Citation**: https://doi.org/10.5194/egusphere-2024-2547-RC1

**References**

[revised manuscript text omitted]

---

## Author Comment (AC2)

**The response to Reviewer #2:**

**Observational Constraints Suggest a Smaller Effective Radiative Forcing from Aerosol-Cloud Interactions**

Park, C., et al

**General Comments**:

The paper uses models and observations to evaluate assumptions made in the determining aerosol-cloud interactions. Specifically, the authors argue that assuming a one-to-one relationship between activation rate of cloud droplet number concentration in response to sulfate aerosol variations and effective radiative forcing by aerosol-cloud interactions (ERFaci) leads to an underestimation of ERFaci. The corroborate this by performing a "perfect-model" validation comparison between climate model "true" ERFaci and that obtained via the aforementioned assumption. They compare observationally constrained ERFaci with previous estimates and conclude that ERFaci may be smaller than previously estimated.

The paper is acceptable with minor revisions (see below). It would be helpful if the authors provided readers with a sense of how widespread the above one-to-one assumption is used in prior studies (e.g., by providing references).

We deeply appreciate the reviewer's thoughtful comments and valuable suggestions. Due to the strict word limits of *ACP Letters*, we have focused on presenting the main findings concisely. Where appropriate, additional details have been added to the Appendix to support for our findings. We kindly ask for your understanding in this regard. Below, we provide specific responses to each comment, highlighted in blue. All "L" references in our responses correspond to line numbers in the revised manuscript with highlighted changes.

**Specific Comments**:

Lines 48-51:"The conventional assumption is that the activation rate has a one-to-one relationship when aerosols convert into cloud droplets and is typically not explicitly incorporated into the estimation process of ERFaci."

Awkward sentence. Please reword. Also, please provide some references where the "conventional assumption" is used.

Thank you for your comment. We agree that the original phrasing was awkward. In L52-55, we revise the sentence to: "In some studies, the activation rate is not explicitly incorporated into the estimation process of ERFaci as it is implicitly assumed to have a one-to-one relationship (e.g. Chen et al., 2014; Christensen et al., 2016; Douglas and L'Ecuyer 2020; Wall et al., 2022, 2023)." This revision provides clarity and includes relevant references.

Line 73: "This ratio, commonly referred to as the activation rate, quantifies the efficiency with which aerosol particles convert into cloud droplets."

Is a constant ratio assumed everywhere? If so, please state this and provide the assumed value.

Thank you for your comment. The activation rate is not assumed to be constant across all conditions. In this sentence, we aimed to define what the activation rate represents rather than imply a fixed value. The activation rate varies based on environmental and aerosol properties, and we address this variability further in the manuscript. To avoid any confusion, we revise the text in L83 to: "This relationship…".

Line 78: "Figure 1".

Please consider using a different color scale. It's not easy to decipher the values when only red is used.

Thank you for your comment. We update the color scheme in Figure 1 to improve clarity and make it easier to distinguish between different values.

Line 87: "The relatively low correlation coefficients observed for…"

Do you mean regression coefficient?

Thank you for pointing this out. We correct "correlation coefficients" to "regression coefficients".

Lines 279-280: "Specifically for sulfate aerosols, it employs bias-corrected observations of total aerosol optical depth in conjunction with…"

Please state where the total aerosol optical depth observations are from.

Thank you for your comment. We specify that the total aerosol optical depth observations are sourced from MODIS satellite data, stating in L312-313: "it employs bias-corrected observations of total aerosol optical depth from the Moderate Resolution Imaging Spectroradiometer (MODIS; Platnick et al., 2015) satellite data…".

**Citation**: https://doi.org/10.5194/egusphere-2024-2547-RC2

**References**

Chen, Y.-C., Christensen, M. W., Stephens, G. L., and Seinfeld, J. H.: Satellite-based estimate of global aerosol–cloud radiative forcing by marine warm clouds, Nature Geosci, 7, 643–646, https://doi.org/10.1038/ngeo2214, 2014.

Christensen, M. W., Chen, Y.-C., and Stephens, G. L.: Aerosol indirect effect dictated by liquid clouds, Journal of Geophysical Research: Atmospheres, 121, 14,636-14,650, https://doi.org/10.1002/2016JD025245, 2016.

Douglas, A. and L'Ecuyer, T.: Quantifying cloud adjustments and the radiative forcing due to aerosol–cloud interactions in satellite observations of warm marine clouds, Atmospheric Chemistry and Physics, 20, 6225–6241, https://doi.org/10.5194/acp-20-6225-2020, 2020.

Wall, C. J., Norris, J. R., Possner, A., McCoy, D. T., McCoy, I. L., and Lutsko, N. J.: Assessing effective radiative forcing from aerosol–cloud interactions over the global ocean, Proceedings of the National Academy of Sciences, 119, e2210481119, https://doi.org/10.1073/pnas.2210481119, 2022.

Wall, C. J., Storelvmo, T., and Possner, A.: Global observations of aerosol indirect effects from marine liquid clouds, Atmospheric Chemistry and Physics, 23, 13125–13141, https://doi.org/10.5194/acp-23-13125-2023, 2023.

---

## Author Comment (AC3)

**The response to Community:**

*This comment is a joint review created as part of EGU's Peer Review Training Workshop 2024. The reviewers were Erin Raif (University of Leeds), Piotr Markuszewski (Institute of Oceanology Polish Academy of Sciences) and Sebastián Mendoza-Téllez (Universidad Nacional Autónoma de México).*

In this paper, the authors used a combination of satellite observations and model reanalysis data to constrain the contribution to effective radiative forcing from aerosol-cloud interactions (ERFaci) in low clouds. In doing so, the authors suggest that previous estimations overestimate ERFaci. They also find that the activation rate of aerosols into cloud droplets must be considered to reduce the uncertainty on effective radiative forcing.

This is an interesting hypothesis with important consequences for the calculation of cloud feedbacks. The data used is comprehensive, the analysis is thorough and the figures are largely clear. However, there is limited discussion placing this work into the context of works that have preceded it, and the importance of the results is not fully explored. Additionally, there are significant issues with the structure of the paper, which does not conform to a typical ACP structure and at times impedes comprehension of the content.

As such, we jointly recommend that this paper be reconsidered after major revisions.

We sincerely appreciate the community's constructive feedback and insightful suggestions. Since our manuscript was submitted to *ACP Letters*, which enforces strict word limits, we have prioritized presenting the main findings concisely. Where necessary, additional details have been included in the Appendix to provide further support for our findings. We kindly request your understanding on this matter. Below, we present specific responses to each comment, highlighted in blue. All "L" references in our responses correspond to line numbers in the revised manuscript with the changes highlighted.

**Major comments**:

1. The authors provide excellent detail to many aspects of their methodology. However, this should generally be contained within the main body of the paper, with only aspects that are unnecessary for comprehension of the methods remaining in the appendices. Additionally, the current structure of the methods/appendix means it is difficult to understand how the individual components fit together – we suggest that the authors **a)** add a short summary at the start of the new methods section to introduce them; **b)** explicitly discuss how their cloud-controlling factor analysis compares to the approach of Wall et al. (2022) and **c)** clearly introduce each data source to help readers familiar with either satellite or reanalysis methods only.

   a) Thank you for your suggestion. Due to the strict word limits of *ACP Letters*, we are unable to provide a more detailed expansion. However, we believe the current structure sufficiently introduces each ERFaci estimation method in Section 2.3, "Perfect-Model Cross Validation", with additional details thoroughly explained in Appendix A4. We appreciate your understanding.

b) Thank you for your comment. We compared our cloud-controlling factor analysis approach with Wall et al. (2022) in Section 2.2, "Observationally Constrained ERFaci", by differentiating between methods that account for activation rate and those that do not. Given the depth of comparison already provided, we believe further discussion in the main text would be redundant.

c) Thank you for your comment. To help readers distinguish between observational and reanalysis data sources, we clarify each data source's origin.

2. While the authors have done an good job of explaining their own approach, there is little discussion as to how this compares to previous work in the field. This leads to two key issues which should be addressed by further discussion of existing literature. **a)** It is difficult to establish from the introduction how ERF_aci is currently estimated, why this approach is limited and how this work differs from those previous approaches. **b)** Reading the discussion and conclusion, it is difficult to establish the relative importance of these new results that the abstract implies. Similarly, it is also difficult to understand the limitations of this approach.

a) Thank you for your comment. I agree that a comprehensive introduction is important. However, due to the strict word limit of *ACP Letters* and the extensive content of our study, we are unable to expand further on the details you mentioned in the introduction. We appreciate your understanding. Additionally, Section 2.2, "Observationally Constrained ERFaci" compares our methodology with previous study and highlights our improvements in subsequent sections.

b) Thank you for your comment. In response, we expand the conclusion section to align more closely with the journal's recommendations. This revision now provides a comprehensive discussion of the uncertainties our study has, ensuring a balanced view of the findings. However, given the strict word limit of ACP Letters, it is challenging to provide more details. We appreciate your understanding regarding these constraints and thank you for your thoughtful review.

3. The results and discussion may be better separated, or at least delineated further with some discussion of each result followed by section for discussion of the results in synthesis.

Thank you for your comment. As mentioned above, we expand the conclusion to include a more comprehensive discussion of our results, including potential ERFaci values based on different data. However, due to the journal's word limit, we streamline this section as much as possible. We appreciate your understanding regarding these constraints.

4. If appropriate, it would be useful to present *p*-values alongside *r*-values throughout the paper to improve the statistical rigour of the findings.

Thank you for pointing this out. We revise the text to include p-values alongside r-values wherever correlation coefficients are reported, and we update captions to indicate "and their associated p-values (p)" for clarity.

**Minor comments:**

Line 34: The authors should consider providing a definition of radiative forcing while still early in the introduction to the paper.

Thank you for your comment. We revise the text to introduce radiative forcing earlier in the introduction. In L29, we now state: "Anthropogenic aerosols impact the Earth's radiation balance at the top of the atmosphere, with this perturbation quantified as radiative forcing (e.g., Boucher et al., 2013; Raghuraman et al., 2021; Kramer et al., 2021).".

Lines 34-36: Not all aerosols act to reduce precipitation and increase cloud liquid water path. For instance, ice-nucleating particles initiate ice formation, which has the opposite effects (though these are unlikely to affect tropical low clouds).

Thank you for your comment. You are correct that not all aerosols serve as CCN; some, such as ice-nucleating particles, initiate ice formation, which can have opposite effects on cloud properties. However, this study focuses specifically on aerosol-cloud interactions within low-level clouds, where ice formation is minimal, and CCN-related interactions are most prominent and well-documented (Christensen et al., 2016; Bellouin et al., 2020; Forster et al., 2021). This generalization is intended to streamline the discussion and maintain relevance to our specific focus on CCN-driven interactions.

Line 49: "the conventional assumption is that…". It would be useful to know who makes this assumption.

Thank you for pointing this out. We revise the sentence to specify in L52: "In some studies, the activation rate is not explicitly incorporated into the estimation process of ERFaci as it is implicitly assumed to have a one-to-one relationship (e.g. Chen et al., 2014; Christensen et al., 2016; Douglas and L'Ecuyer 2020; Wall et al., 2022, 2023).".

Line 51: Is there a reference that justifies the "one-to-one aerosol cloud relationship" argument? [see also comments re. lines 70 and 80]

Thank you for your comment. In response, we revise the sentence to specify in L52: "In some studies, the activation rate is not explicitly incorporated into the estimation process of ERFaci as it is implicitly assumed to have a one-to-one relationship (e.g. Chen et al., 2014; Christensen et al., 2016; Douglas and L'Ecuyer 2020; Wall et al., 2022, 2023).".

Line 66: Can the authors here explicitly summarise the main focus of the "story" by presenting a research hypothesis or clear research questions?

Thank you for your comment. We appreciate the suggestion to clarify the focus. In the introduction, we highlighted the considerable uncertainties in estimating ERFaci in L41 and presented the main focus of the study in L45-62, starting with "Estimating the ERFaci, especially in low-level clouds…". Additionally, we introduced the specific aerosol proxy we focus on. We believe this structure effectively conveys our research questions and objectives.

Lines 70-74: This goes some way to answering the comment on line 51, so should probably be moved to the introduction. However, further expansion may also be helpful for the reader – why is the assumption of a 1-1 ratio wrong?

Thank you for your comment. The assumption of a 1:1 ratio implies a direct, linear relationship between the variables in question—meaning that any increase in one variable would yield an equal proportional increase in the other. However, this assumption is often oversimplified in climate and atmospheric sciences because numerous factors influence the interaction between these variables. We clarify this in the main text with the addition in L92: "…underscoring that not all $SO_4$ in the atmosphere are converted into cloud droplets.".

Line 80: The relative strength of the relationship in different regions is very interesting. In most regions, the relationship is proportional but not one-to-one. Could the authors clarify why this might not be expected, as SO4 is only a subset of aerosol?

Thank you for pointing this out. We clarify this in the main text with the addition in L98: "This variation may be attributed to differences in local environmental conditions and the role of aerosols in which these clouds occur (e.g. Douglas and L'Ecuyer, 2019, 2020)."

Figure 1: We think the interpretation of the plot is good. Can correlation coefficients of 0.4 be described as highly linear?

Thank you for pointing this out. To avoid confusion and improve clarity, we remove the correlation coefficient from Figure 1 and Figure S1, as it was not ideal for visually representing the activation rate. Regarding linearity, we previously used diagonal hatching to indicate areas where correlation coefficients exceeded 0.4, as we considered this value relatively high for each grid point.

Line 113: Eq. 1 implies there are ten states that LWP can be in. Can the authors clarify what they mean by this, and perhaps briefly detail them?

Thank you for pointing this out. Initially, we considered using liquid water path (LWP) bins to further specify cloud properties based on LWP. However, to maintain consistency and directly compare our results with those of Wall et al. (2022), which do not constrain LWP, we decide to

exclude LWP in our equations. Even without this constraint, our results remain consistent, supporting the robustness of our findings.

Figure 2: The difference in meanings between panels (b) and (c) and panels (d) and (e) could be clarified by adding description to the colourbar adjacent to plot (c).

Thank you for your suggestion. We considered adding descriptions to the colorbar for clarity; however, we believe that the current labels "With activation" and "Without activation" more directly convey the differences between the panels.

Line 179-188: In Figures 3a and 3c, the authors clearly show the ensemble ERF_aci is improved by considering activation rate. However, there are similar absolute numbers of models which perform well regardless of the treatment. Are these the same models in each case, and if so, is there an indicator as to when considering activation rate is important to capture ERF_aci and when it is not?

Thank you for your comment. Upon review, we found that the models aligning well along the y=x line are indeed the same, indicating they consistently perform well in simulating aerosol-cloud interactions. However, as this detail falls slightly outside the main focus of our study, we opt not to include it in the main text. We appreciate your insight nonetheless.

Line 200-201: "…our estimates offer further evidence to support estimates on the lower end of [the WCRP's] range". This seems to contradict Fig 4, where the red bars indicating ERFaci_obs have the largest values. Do the authors mean to say less negative?

Thank you for pointing that out. We revise the text to "higher end" and add "(less negative)" for additional clarification.

Line 202: Could the authors clarify what a "top-down" approach is and how it differs from their analysis?

Thank you for your comment. We understand the concern and, to ensure clarity, we decide to replace the term "top-down" in the main text with "recent".

Line 220: What threshold was chosen for models to fall into the GOOD HIST category?

Thank you for your comment. We add further explanation in Appendix 1.4 at L364: "For analysis, we select the 15 models with the lowest GOOD HIST indices (Table S1).".

Figure 5: We assumed that, like other plots, the solid dots referred to values obtained when activation rate was considered. However, it would be useful to specify this in the caption. Additionally, there is no colourbar – we think this is because the colours correspond exactly to

the x-axis. If this this the case, consider removing the colours as it implies an extra variable (such as each colour representing a different model) and the yellow unfilled circles are difficult to see.

Thank you for your comment. For clarity, we add "when the activation rate is accounted for" to the caption. We choose to retain the current color scheme to visually highlight how our observationally constrained ERFaci values compare with model outputs, emphasizing the relative differences. We hope this provides a clearer comparison, though we appreciate your suggestion.

Line 247: For readers who are reading the paper non-linearly, consider specifying the degree to which the influence of aerosols may be less substantial than assumed.

Thank you for your comment. We add our estimates alongside the IPCC estimate in the text at L265: "(e.g., -0.93 ± 0.7 W m$^{-2}$ in IPCC AR6, 90% confidence)".

Line 335: Does the choice of a 50 year period remove interannual variability or reduce the influence of it? And if so, relative to what?

Thank you for pointing this out. The choice of a 50-year period helps reduce the influence of interannual variability by averaging over a longer timescale, which smooths out year-to-year fluctuations that might otherwise disproportionately affect shorter-term analyses. We revise the text in L373 to say "…to minimize the influence of interannual variability" for clarity.

Line 335-337: Was there a specific reason why the 13 and 9 models were chosen for SO4 and AI respectively?

Thank you for your comment. The choice of 13 models for $SO_4$ and 9 models for AI was based on model availability within CMIP6. We add the phrase "Due to the limited availability of models for aerosol proxies…"

Line 370: To make this clearer for the reader, consider ending this sentence with e.g. "in this case, SO4 concentrations or AI".

Thank you for pointing this out. To improve clarity, we add the following sentence at L414-416: "Specifically, we consider either the natural logarithm of $SO_4$ at 925 hPa from the MERRA-2 reanalysis or the natural logarithm of the AI from MODIS.".

Line 403: What is the 1pctCO2 scenario?

Thank you for your comment. The 1pctCO2 scenario represents an experiment in which $CO_2$ concentrations increase by 1% per year, as provided by CMIP6. I revise the text to clarify this, now stating at L461: "...where $\alpha_{1pctCO_2}$ represents the low-level cloud feedback, derived from the 1% $CO_2$ increase per year (1pctCO$_2$) scenario...".

Line 437: Consider using "more negative" rather than larger.

Thank you for your suggestion. In this context, the values represent multiples, so we believe "larger" is appropriate in this sentence to convey the intended meaning accurately.

Line 457: Is there any literature to back up the assertion that the polar oceans will not contribute largely to the ERF_aci?

Thank you for pointing this out. Assessing aerosol-cloud interactions in polar regions remains challenging due to significant cloud model uncertainties, spatial and temporal observation limitations, and the difficulty of obtaining some types of remote sensing information at high latitudes. These limitations make it difficult to quantify the polar oceans' contribution to ERFaci reliably.

Instead, we expand our study domain to cover the area between 60°S and 60°N, and our conclusions remain consistent within this extended area. We believe that excluding polar regions does not significantly impact our conclusions, as sulfate mass concentrations are primarily concentrated in major industrial regions like East Asia and North America, which are well represented within our study domain.

Additionally, to clarify, we adjust the text to avoid potential confusion regarding polar region contributions. We now highlight the extrapolation approach using CMIP6 single-forcing experiments, where we apply a scalar multiplier to the domain-average value to estimate a global-average value, as explained in Appendix A6 and illustrated in Figure A3.

Line 477: Consider a section title that is more specific than "Uncertainty".

Thank you for your comment. We revise it to "**A7 Uncertainty from ERFaci_obs estimation**" to provide better specificity.

Table A1: Please add more detail to the caption, such as what the circles mean and brief redefinitions of the variables.

Thank you for pointing this out. We revise the caption to clarify: "CMIP6 models used in the analysis are represented, with each circle indicating the availability of data for a given model. $\Delta\ln(SO_4)$ and $\Delta\ln(AI)$ represent changes in sulfate mass concentration and aerosol index, respectively, from present day to pre-industrial levels on a natural logarithmic scale. ERFaci_true refers to ERFaci derived from single-forcing (aerosol-only) experiments, while ERFaci_SC17 is calculated using the method from Soden and Chung (2017). ERFaci_est (SO$_4$) and ERFaci_est (AI) denote estimated ERFaci values based on simplified version of Eq. (1) and Eq. (2) in the

CMIP6 models (Appendix A4). The GOOD HIST index represents the absolute difference in global-mean historical warming compared to observations (Appendix A1.4).".

**Technical corrections**:

Throughout: the authors should consider when to italicise and when to romanise variables and subscripts in equations, which is discussed in the ACP guidelines ([https://www.atmospheric-chemistry-and-physics.net/submission.html#math](https://www.atmospheric-chemistry-and-physics.net/submission.html#math))

Thank you for pointing this out. We correct the formatting of variables and subscripts in equations to adhere to the ACP guidelines.

Line 14: replace "it is assumed" with "assume".

Thank you for pointing that out. We revise the text, "While some studies assume that…".

Line 39: ERFaci has not yet been defined in the text, just in the abstract, so this should precede the abbreviation.

Thank you for pointing out. We revise the text to include the full term before introducing the abbreviation "ERFaci".

Figure 1, line 794: add the word "hatching" after diagonal.

Thank you for your comment. However, for improved visibility, we remove the diagonal hatching from Figure 1 and Figure S1.

Figure 1, line 796: "stippling" might be my new favourite word!

Thank you for your comment.

Figure 1, line 797: "Stduent's" should be "Student's".

Thank you for pointing out. We correct it.

Line 326: The sentence beginning "So, the models…" is a clause that doesn't form a full sentence. Consider a change such as "This suggests that the models…"

Thank you for your suggestion. To avoid multiple use of "that" we instead revise it to "This suggests the models that…".

Line 491: delete the second instance of the word "the" in the phrase "hence **C**ii represents the diagonal components of the **C**".

Thank you for pointing out. We correct it.

Lines 521 and 530: these are quite unwieldy and should probably be standalone equations.

Thank you for your suggestion. We revise these expressions as standalone equations to improve readability.

**Citation**: https://doi.org/10.5194/egusphere-2024-2547-CC1

**References**

Bellouin, N., Quaas, J., Gryspeerdt, E., Kinne, S., Stier, P., Watson-Parris, D., Boucher, O., Carslaw, K. S., Christensen, M., Daniau, A.-L., Dufresne, J.-L., Feingold, G., Fiedler, S., Forster, P., Gettelman, A., Haywood, J. M., Lohmann, U., Malavelle, F., Mauritsen, T., McCoy, D. T., Myhre, G., Mülmenstädt, J., Neubauer, D., Possner, A., Rugenstein, M., Sato, Y., Schulz, M., Schwartz, S. E., Sourdeval, O., Storelvmo, T., Toll, V., Winker, D., and Stevens, B.: Bounding Global Aerosol Radiative Forcing of Climate Change, Reviews of Geophysics, 58, e2019RG000660, https://doi.org/10.1029/2019RG000660, 2020.

Boucher, O., Randall, D., Artaxo, P., Bretherton, C., Feingold, G., Forster, P., Kerminen, V.-M., Kondo, Y., Liao, H., Lohmann, U., Rasch, P., Satheesh, S.K., Sherwood, S., Stevens, B., and Zhang, X.Y.: Clouds and aerosols, in: Climate Change 2013: The Physical Science Basis. Contribution of Working Group I to the Fifth Assessment Report of the Intergovernmental Panel on Climate Change, edited by: Stocker, T.F., Qin, D., Plattner, G.-K., Tignor, M., Allen, S.K., Doschung, J., Nauels, A., Xia, Y., Bex, V., and Midgley, P.M., Cambridge University Press, Cambridge, UK and New York, NY, USA, 571-657, https://doi.org/10.1017/CBO9781107415324.016, 2013.

Chen, Y.-C., Christensen, M. W., Stephens, G. L., and Seinfeld, J. H.: Satellite-based estimate of global aerosol–cloud radiative forcing by marine warm clouds, Nature Geosci, 7, 643–646, https://doi.org/10.1038/ngeo2214, 2014.

Christensen, M. W., Chen, Y.-C., and Stephens, G. L.: Aerosol indirect effect dictated by liquid clouds, Journal of Geophysical Research: Atmospheres, 121, 14,636-14,650, https://doi.org/10.1002/2016JD025245, 2016.

Douglas, A. and L'Ecuyer, T.: Quantifying cloud adjustments and the radiative forcing due to aerosol–cloud interactions in satellite observations of warm marine clouds, Atmospheric Chemistry and Physics, 20, 6225–6241, https://doi.org/10.5194/acp-20-6225-2020, 2020.

Douglas, A. and L'Ecuyer, T.: Quantifying variations in shortwave aerosol–cloud–radiation interactions using local meteorology and cloud state constraints, Atmospheric Chemistry and Physics, 19, 6251–6268, https://doi.org/10.5194/acp-19-6251-2019, 2019.

Forster, P., Storelvmo, T., Armour, K., Collins, W., Dufresne, J.-L., Frame, D., Lunt, D.J., Mauritsen, T., Palmer, M.D., Watanabe, M., Wild, M., and Zhang, H.: The Earth's Energy Budget, Climate Feedbacks, and Climate Sensitivity, in: Climate Change 2021: The Physical Science Basis. Contribution of Working Group I to the Sixth Assessment Report of the Intergovernmental Panel on Climate Change, edited by: Masson-Delmotte, V., Zhai, P., Pirani, A., Connors, S.L., Péan, C., Berger, S., Caud, N., Chen, Y., Goldfarb, L., Gomis, M.I., Huang, M., Leitzell, K., Lonnoy, E., Matthews, J.B.R., Maycock, T.K., Waterfield, T., Yelekçi, O., Yu,

R., and Zhou, B., Cambridge University Press, Cambridge, UK and New York, NY, USA, 923–1054, https://doi.org/10.1017/9781009157896.009, 2021.

Kramer, R. J., He, H., Soden, B. J., Oreopoulos, L., Myhre, G., Forster, P. M., and Smith, C. J.: Observational Evidence of Increasing Global Radiative Forcing, Geophysical Research Letters, 48, e2020GL091585, https://doi.org/10.1029/2020GL091585, 2021.

Raghuraman, S. P., Paynter, D., and Ramaswamy, V.: Anthropogenic forcing and response yield observed positive trend in Earth's energy imbalance, Nat Commun, 12, 4577, https://doi.org/10.1038/s41467-021-24544-4, 2021.

Soden, B. and Chung, E.-S.: The Large-Scale Dynamical Response of Clouds to Aerosol Forcing, Journal of Climate, 30, 8783–8794, https://doi.org/10.1175/JCLI-D-17-0050.1, 2017.

Wall, C. J., Norris, J. R., Possner, A., McCoy, D. T., McCoy, I. L., and Lutsko, N. J.: Assessing effective radiative forcing from aerosol–cloud interactions over the global ocean, Proceedings of the National Academy of Sciences, 119, e2210481119, https://doi.org/10.1073/pnas.2210481119, 2022.

Wall, C. J., Storelvmo, T., and Possner, A.: Global observations of aerosol indirect effects from marine liquid clouds, Atmospheric Chemistry and Physics, 23, 13125–13141, https://doi.org/10.5194/acp-23-13125-2023, 2023.

---

## Author Response (AR2)

**The response to Reviewer #1:**

We sincerely thank the reviewer for their constructive comments and valuable suggestions. Below, we provide specific responses to each comment, highlighted in blue.

1. I had a concern about the layout and structure of the article with most of the technical information (data and methods) in the appendix. I am still confused and compared to the letter format of other journals, there is more information on data and methods in them. I am not familiar with ACP letters, so I leave it to the editor to decide if there is enough information in the main article.

Thank you for your comment. We understand your concern regarding the placement of methodological details in the appendix. In ACP Letters, due to the strict word limit (a maximum of 2,500 words), many published papers present data and methods in the appendix rather than in the main text. For reference, we have cited similar examples below.

Given these limitations, we aimed to balance conciseness with methodological clarity. Our manuscript currently reaches 2,474 words (excluding references), which limits the extent of methodological discussion we can include in the main text. To ensure transparency while adhering to the journal's format, we have provided key methodological details in the appendix while keeping the main text focused on the primary findings and their implications.

While we recognize that some letter-format journals allow for more methodological content in the main text due to a higher word limit, we have followed ACP Letters' formatting conventions. We defer to the editor's judgment on whether the current level of detail is appropriate for this journal and are open to any recommendations for adjustments.

References

Diamond, M. S.: Detection of large-scale cloud microphysical changes within a major shipping corridor after implementation of the International Maritime Organization 2020 fuel sulfur regulations, Atmospheric Chemistry and Physics, 23, 8259–8269, https://doi.org/10.5194/acp-23-8259-2023, 2023.

Ploeger, F., Birner, T., Charlesworth, E., Konopka, P., and Müller, R.: Moist bias in the Pacific upper troposphere and lower stratosphere (UTLS) in climate models affects regional circulation patterns, Atmospheric Chemistry and Physics, 24, 2033–2043, https://doi.org/10.5194/acp-24-2033-2024, 2024.

Teixeira, J., Wilson, R. C., and Thrastarson, H. T.: Direct observational evidence from space of the effect of $CO_2$ increase on longwave spectral radiances: the unique role of high-spectralresolution measurements, Atmospheric Chemistry and Physics, 24, 6375–6383, https://doi.org/10.5194/acp-24-6375-2024, 2024.

2. I am still confused about the extension to the whole domain. It is still based on a hypothesis and the article is not very clear about this (or I miss the sentence). I agree with the authors that sulphate mass concentrations are mainly concentrated in major industrial regions, but the Arctic regions have shown that the effect of aerosols acting as CCN is particularly efficient in the Arctic (up to 8 times more efficient compared to mid-latitudes, see coopman et al., 2018).
Coopman, Q., Garrett, T. J., Finch, D. P., & Riedi, J. (2018). High sensitivity of arctic liquid clouds to long-range anthropogenic aerosol transport. Geophysical Research Letters, 45, 372–381. https://doi.org/10.1002/2017GL075795

Thank you for your insightful comment. We understand your concern regarding the extrapolation of our domain-average ERFaci to a global estimate. As noted in the manuscript, limitations in satellite observations prevent us from obtaining continuous and reliable data over land and polar regions (Jia et al., 2019; Gryspeerdt et al., 2022; Jia and Quaas, 2023). Under these circumstances, to account for the global-average ERFaci from our domain-average ERFaci, we use CMIP6 single-forcing experiments to derive a scalar multiplier ($\gamma$) based on the ratio of the multi-model mean of global-average ERFaci_true to domain-average ERFaci_true. Importantly, this approach inherently accounts for regional variations, as $\gamma$ is derived from a multi-model estimate that includes polar aerosol-cloud interactions. Our analysis finds $\gamma = 0.86$, with a strong linear correlation ($r = 0.92$, $p < 0.001$), ensuring the robustness of our extrapolation methodology. This approach is detailed in Appendix A6 and illustrated in Figure A3.

To further validate our methodology, we perform a sensitivity test using observational dataset following Wall et al. (2022). In this alternative method, we assume that the albedo change associated with ERFaci is approximately uniform across the study domain and the entire globe. Under this assumption, $\gamma$ is estimated as the ratio of global-mean to domain-mean insolation, yielding a central estimate of 0.92. Notably, this estimate is highly consistent with our model-derived value of 0.86, reinforcing the robustness of our extrapolation approach. Given this consistency, we adopt $\gamma = 0.86$ in this study.

We acknowledge that aerosol-cloud interactions in the Arctic may be more efficient per unit aerosol mass due to unique atmospheric conditions (e.g., pristine environment, polar night/day cycles). If the Arctic's high aerosol-cloud interaction efficiency were fully considered, our global ERFaci estimate could potentially be more negative. However, the efficiency estimates in Coopman et al. (2018) are based on a dataset with limited temporal (March to September, 2005–2010) and spatial coverage (north of 65°N over the ocean). Thus, incorporating these effects with arbitrary weighting (e.g., 2 to 8 times higher efficiency in the Arctic than in the mid-latitudes) could introduce additional uncertainties in our analysis. Nonetheless, we recognize the importance of Arctic aerosol-cloud interactions and their potential influence on global-mean ERFaci estimates.

To address your concern, we have revised the manuscript to clarify the potential uncertainties associated with Arctic aerosol-cloud interactions and their role in our extrapolation methodology. Specifically, we now state in Appendix A6:

"Additionally, following the approach of Wall et al. (2022), we conduct a sensitivity test for $\gamma$ without relying on climate model results. In this alternative method, we assume that the albedo change associated with ERFaci is approximately uniform across the study domain and the entire globe. Under this assumption, $\gamma$ is approximated as the ratio of global-mean insolation to domain-mean insolation, yielding a central estimate of 0.92. Notably, this value is highly consistent with our model-derived estimate of 0.86, supporting the robustness of our extrapolation approach. Given this consistency, we adopt $\gamma = 0.86$ in this study.

Even though our study domain captures the primary anthropogenic aerosol sources, particularly near major industrial regions in Eurasia and North America, and our multi-model mean extrapolation inherently accounts for aerosol-cloud interactions outside our domain, recent studies have highlighted their significant influence in polar regions (e.g., Coopman et al., 2018). Aerosol-induced cloud property changes in the Arctic may be more efficient per unit aerosol mass than at mid-latitudes due to the greater susceptibility of Arctic clouds to aerosols. Incorporating these effects could lead to a more negative global-mean ERFaci estimate. The role of Arctic aerosol-cloud interactions warrants further investigation, and future research incorporating more comprehensive observational constraints would be valuable."

Additionally, we have revised line 144 of the manuscript to explicitly state how our global estimate is extrapolated:
"To estimate global-average ERFaci_obs from our domain-average ERFaci_obs, we multiply our domain estimate by a scalar multiplier, $\gamma$, which represents the ratio of multi-model mean of global-average ERFaci_true to domain-average ERFaci_true (Appendix A6)."

References

Coopman, Q., Garrett, T. J., Finch, D. P., and Riedi, J.: High Sensitivity of Arctic Liquid Clouds to Long-Range Anthropogenic Aerosol Transport, Geophysical Research Letters, 45, 372–381, https://doi.org/10.1002/2017GL075795, 2018.

Gryspeerdt, E., McCoy, D. T., Crosbie, E., Moore, R. H., Nott, G. J., Painemal, D., Small-Griswold, J., Sorooshian, A., and Ziemba, L.: The impact of sampling strategy on the cloud droplet number concentration estimated from satellite data, Atmospheric Measurement Techniques, 15, 3875–3892, https://doi.org/10.5194/amt-15-3875-2022, 2022.

Jia, H. and Quaas, J.: Nonlinearity of the cloud response postpones climate penalty of mitigating air pollution in polluted regions, Nat. Clim. Chang., 13, 943–950, https://doi.org/10.1038/s41558-023-01775-5, 2023.

Jia, H., Ma, X., Quaas, J., Yin, Y., and Qiu, T.: Is positive correlation between cloud droplet effective radius and aerosol optical depth over land due to retrieval artifacts or real physical processes?, Atmospheric Chemistry and Physics, 19, 8879–8896, https://doi.org/10.5194/acp-19-8879-2019, 2019.

Wall, C. J., Norris, J. R., Possner, A., McCoy, D. T., McCoy, I. L., and Lutsko, N. J.: Assessing effective radiative forcing from aerosol–cloud interactions over the global ocean, Proceedings of the National Academy of Sciences, 119, e2210481119, https://doi.org/10.1073/pnas.2210481119, 2022.

3. I disagree with the authors about the convention of referring to SO42- as SO4 and think it should be changed throughout the text. I agree that SO4 could be clear to readers, but I do not think it is correct to write it in this way. I will follow the editor decision with this matter.

Thank you for your concern. Based on your suggestion, we have decided to use "$SO_4^{2-}$" instead of "$SO_4$" throughout the manuscript. We believe this revision enhances readability and aligns with conventional chemical notation. We appreciate your feedback.

4. I still have concerns about using only SO4 as an aerosol. On line 110 of the manuscript, the authors state that it is 64% smaller with activation than without. Taking into account other aerosols might potentially diminish the difference. The information with the AI is more impactful in this regards.

Thank you for your comment. We understand your concern regarding the use of $SO_4$ as the primary aerosol proxy. Our decision is based on its well-documented dominant role in aerosol-cloud interactions and its strong correlation with cloud droplet number concentrations compared to other aerosol types (Charlson et al., 1992; Stevens, 2015; McCoy et al., 2018). To provide a broader perspective and strengthen our analysis, we also included the Aerosol Index (AI) as an additional aerosol proxy. The consistency in ERFaci estimates derived from both proxies highlights the robustness of our findings.

Sulfate has been widely used in previous studies as a primary proxy for aerosol-cloud interactions, particularly in the context of radiative forcing (e.g., Wall et al., 2022; Gryspeerdt et al., 2023). Additionally, $SO_4$ concentrations at 925 hPa offer a more direct representation of CCN availability near the cloud base, which is crucial for assessing cloud microphysical processes (Painemal et al., 2017). In contrast, AI from MODIS represents a column-integrated aerosol quantity that does not account for vertical aerosol distribution, making it less precise in capturing aerosol-cloud interactions. Nevertheless, its inclusion in our study provides a complementary perspective on aerosol proxy.

While considering additional aerosol types could further refine the analysis, $SO_4$ remains widely used and physically relevant proxy for estimating ERFaci. Given the consistency of ERFaci

estimates across both $SO_4$ and AI, we are confident that our approach provides a robust and comprehensive assessment. We appreciate your feedback and believe that the use of these two proxies strengthens the reliability of our findings.

To further clarify this point, we have revised line 50 of the manuscript to emphasize the dominant role of sulfate in ERFaci:

"Sulfate aerosol is recognized as a dominant contributor to ERFaci as well as cloud droplet formation, alongside other aerosol types such as black carbon, organic carbon, sea salt, and dust (Charlson et al., 1992; Stevens, 2015; McCoy et al., 2018)."

Additionally, we have highlighted the column-integrated feature of AI in Appendix 1.3: "However, it is important to note that since AI provides column-integrated quantities and does not account for the vertical profile, it may not accurately capture aerosol concentrations in low-level clouds, which are the focus of our study."

References

Charlson, R. J., Schwartz, S. E., Hales, J. M., Cess, R. D., Coakley, J. A., Hansen, J. E., and Hofmann, D. J.: Climate Forcing by Anthropogenic Aerosols, Science, 255, 423–430, https://doi.org/10.1126/science.255.5043.423, 1992.

Gryspeerdt, E., Povey, A. C., Grainger, R. G., Hasekamp, O., Hsu, N. C., Mulcahy, J. P., Sayer, A. M., and Sorooshian, A.: Uncertainty in aerosol–cloud radiative forcing is driven by clean conditions, Atmospheric Chemistry and Physics, 23, 4115–4122, https://doi.org/10.5194/acp-23-4115-2023, 2023.

McCoy, D. T., Bender, F. A.-M., Grosvenor, D. P., Mohrmann, J. K., Hartmann, D. L., Wood, R., and Field, P. R.: Predicting decadal trends in cloud droplet number concentration using reanalysis and satellite data, Atmospheric Chemistry and Physics, 18, 2035–2047, https://doi.org/10.5194/acp-18-2035-2018, 2018.

Painemal, D., Chiu, J.-Y. C., Minnis, P., Yost, C., Zhou, X., Cadeddu, M., Eloranta, E., Lewis, E. R., Ferrare, R., and Kollias, P.: Aerosol and cloud microphysics covariability in the northeast Pacific boundary layer estimated with ship-based and satellite remote sensing observations, Journal of Geophysical Research: Atmospheres, 122, 2403–2418, https://doi.org/10.1002/2016JD025771, 2017.

Stevens, B.: Rethinking the Lower Bound on Aerosol Radiative Forcing, Journal of Climate, 28, 4794–4819, https://doi.org/10.1175/JCLI-D-14-00656.1, 2015.

Wall, C. J., Norris, J. R., Possner, A., McCoy, D. T., McCoy, I. L., and Lutsko, N. J.: Assessing effective radiative forcing from aerosol–cloud interactions over the global ocean, Proceedings of the National Academy of Sciences, 119, e2210481119, https://doi.org/10.1073/pnas.2210481119, 2022.

5. Some of the texts in the appendix are referenced in the main text : Appendix A, A1.2, A1.1 for example.

Thank you for pointing this out. We intentionally reference specific sections of the appendix, such as Appendix A3 and A4, in the main text to provide additional methodological details while keeping the manuscript within the word limit.

However, if your concern is that certain sections of the appendix, such as Appendix A1.1 and A1.2, which primarily contain data details rather than methodological explanations, are not explicitly referenced in the main text, this was a deliberate decision to enhance readability and avoid unnecessary redundancy. We sincerely appreciate your thoughtful feedback and believe that this approach does not compromise the clarity or comprehension of the main text.

---

## Author Response (AR3)

**The response to Editor:**

We sincerely appreciate the editor's time, consideration, and valuable suggestions in reviewing our manuscript. Below, we provide specific responses to each comment, with our revisions highlighted in blue.

1. Thank you for your careful responses to the constructive reviews. I am now willing to forward your provocative manuscript for publication in ACP. However, I have two minor requests. First is that the abbreviation ERFaci be reintroduced in the Conclusions.

Thank you for your comment. We have now reintroduced the abbreviation ERFaci in the Conclusions, specifically in L184: "Our study offers critical insights into the quantification of the effective radiative forcing from aerosol-cloud interactions (ERFaci)…".

2. The second is that the results for the sensitivity shown in Figure 1 be summarized numerically in the abstract, body, conclusions. Non-linearity is central to the paper's arguments, and this can be quantified for example by a global mean oceanic value to give the reader a sense of the magnitude of the difference. It looks to be roughly 0.5.

Thank you for this insightful comment. We have now incorporated numerical summaries of the global mean activation rate throughout the manuscript. To ensure consistency with the ERFaci estimation, we followed the methodology outlined in Appendix A6 and included the associated uncertainties based on Appendix A8. Below are the specific revisions:

Abstract (L14): "Our analysis estimates a global mean activation rate of $0.35 \pm 0.17$ (90% confidence)…".

Section 2.1 (L65): "On a global scale, the mean activation rate is 0.35, indicating that sulfate aerosol activation is less efficient than a one-to-one conversion.".

Section 2.1 (L71): "with a global mean of 0.21 (Fig. S1).".

Conclusion (L187): "…estimated globally at $0.35 \pm 0.17$ for $SO_4^{2-}$ and $0.21 \pm 0.23$ for AI (90% confidence)…".

Appendix A6 (L451): "We also apply this scalar multiplier to extrapolate the global mean activation rate, as variations in $N_d$ in single-forcing (aerosol-only) experiments primarily result

from changes in aerosol concentrations. This extrapolation remains consistent with the ratio of global mean ERFaci_obs calculated with and without accounting for activation rate, suggesting a global mean activation rate of 0.37 for $SO_4^{2-}$ and 0.21 for AI.".

Appendix A8 (L522): "The uncertainty in the activation rate is calculated in a similar manner, but it arises from the regression coefficient of $\partial \ln(N_d) / \partial \ln(X)$ and the extrapolation of the global activation rate. The term $\delta$ is computed following Eq. (A9) but excluding $[\Delta \ln(X)]$ and using $\ln(N_d)'$ in place of CRE_lcld'. To estimate the uncertainty in spatially averaged regression coefficients for the activation rate, we employ Eq. (A11). Consequently, the overall 90% CI for the global activation rate is given by

$$\text{Activation rate, global} \pm \sqrt{([\gamma]\Delta_{obs})^2 + \Delta_\gamma^2}. \qquad \text{(A14)}$$ ".